# TOWARDS GUARANTEES FOR PARAMETER ISOLATION IN CONTINUAL LEARNING

## ABSTRACT

Deep learning has proved to be a successful paradigm for solving many challenges in machine learning. However, deep neural networks fail when trained sequentially on multiple tasks, a shortcoming known as *catastrophic forgetting* in the *continual learning* literature. Despite a recent flourish of learning algorithms successfully addressing this problem, we find that provable guarantees against catastrophic forgetting are lacking. In this work, we study the relationship between learning and forgetting by looking at the geometry of neural networks' loss landscape. We offer a unifying perspective on a family of continual learning algorithms, namely methods based on *parameter isolation*, and we establish guarantees on catastrophic forgetting for some of them.

## 1 INTRODUCTION AND MOTIVATION

Statistical models based on deep neural networks are trusted with ever more complex tasks in real-world applications. In real-world environments the ability to continually and rapidly learn new behaviors is crucial. It is therefore worthwhile to understand how deep neural networks store and integrate new information. In this paper, we want to study neural networks in the *continual learning* setting, where the input to the learning algorithm is a *data stream*. In this setting, it has been observed that training neural networks on new data often severely degrades the performance on old data, a phenomenon termed *catastrophic forgetting* (McCloskey & Cohen, 1989), which we will often simply refer to as *forgetting* herefter.

Generally speaking, continual learning algorithms address catastrophic forgetting by leveraging storage external to the network and imposing constraints (implicit, or explicit) that ensure the network does not stray too far off from the prior tasks when a new task is given. The storage gets updated with each new learning task and its specific contents depend on the algorithm: typical examples include vectors in the parameter space (network parameters or gradients), input samples, or neural activities. We review the main trends in the literature in the related work section (Section 2).

The last couple of years have witnessed significant progress in continual learning (De Lange et al., 2021; Khetarpal et al., 2022). However, most of this progress has largely been empirical in nature, without any rigorous theoretical understanding of the successful strategies. As a consequence, although the algorithms proposed are shown to reduce catastrophic forgetting on several benchmarks, there is no guarantee that they will achieve the same low forgetting rate in other contexts, and, in general, it is hard to obtain a grip over the inherent limitations of the algorithms. Arguably, these characteristics are necessary to make such algorithms reliable in the wild.

In this study, we focus on theoretically understanding a family of continual learning methods based on *parameter isolation* strategies. We propose a unified framework for these algorithms from a loss landscape perspective. More specifically, we approximate each task's loss around the optimum and derive a constraint on the parameter update to prevent forgetting. We show that many existing algorithms are special cases of our framework.

Moreover, our work complements previous studies of catastrophic forgetting for another family of continual learning algorithms, namely *regularisation-based* methods. Our results suggest that these two different strategies are more similar than what initially thought.

We begin by introducing formally the continual learning problem. In Section 2 we review the literature most relevant for this study. In Section 3 we introduce our framework through an analysis of the

tasks' loss geometry , and in Section 4 we show that several existing parameter isolation algorithms can be described within this framework and we establish guarantees for two such algorithms. Finally, Section 5 is dedicated to the empirical validation of our theoretical findings.

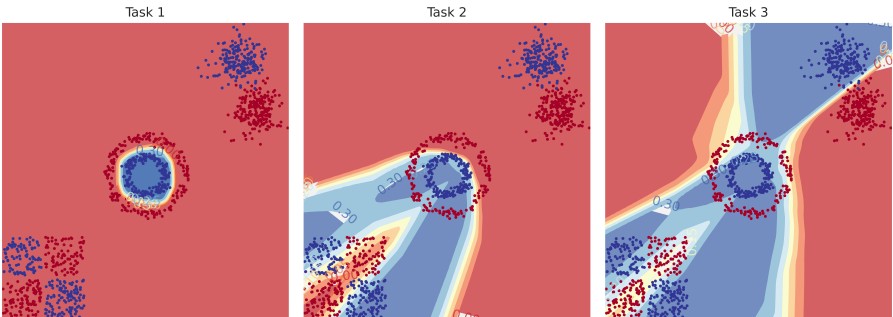

Figure 1: Illustration of catastrophic forgetting on a toy challenge. An MLP is trained *sequentially* on 3 geometric tasks with 2 classes (in red and blue). The first task occupies the center of the space, the second one the bottom left and the last one the top right corner. Each column shows the network's decision boundary after learning the corresponding task: after the first task (left), the network can perfectly discriminate between the two classes, however, the decision boundary shifts with the second (middle) and third (right) tasks and the network forgets the first task entirely. The multi-task solution is shown in the Appendix 4.

## 1.1 PRELIMINARIES

The term continual learning generally refers to a learning problem in which the data is accessed sequentially. It is typically assumed that the data stream is *locally i.i.d.*, thereby it can be written as a sequence of tasks $\mathcal{T}_1, \ldots, \mathcal{T}_T$. In this case, the learning algorithm cannot access examples of previous or future tasks. We restrict our study to the supervised learning case as it is the most explored in the literature. Each task $\mathcal{T}_t$ comes with a dataset $D_t$, which in the supervised learning scenario consists of a set of pairs $(x, y) \in \mathcal{X}_t \times \mathcal{Y}_t$. Another typical assumption is that the input space is shared across tasks, i.e. $\mathcal{X}_t = \mathcal{X} \ \forall t \in [1, \ldots, T]$.

We define the neural network with a set of parameters $\Theta_t$. Hereafter, we assume w.l.o.g. $\Theta_t \subseteq \Theta \ \forall \ t$ and use $\Theta \subseteq \mathbb{R}^P$ as parameter space (we elaborate on this choice in Appendix A). We use $\boldsymbol{\theta}$ to refer to a generic vector in the parameter space and $\boldsymbol{\theta}_t$ to the value of the parameters found by training on the $t$-th task. Normally, we index the current task with $t$ and any old task with $o$. The neural network implements a function $f(\cdot; \boldsymbol{\theta}) : \mathcal{X} \to \mathcal{Y}_t$. We write $l_t(x, y, \boldsymbol{\theta})$ for the $t$-th task loss function, and we refer to the average loss over the dataset with $\mathcal{L}_t(\boldsymbol{\theta})$. We shorten the loss measured at the end of training $\mathcal{L}_t(\boldsymbol{\theta}_t)$ to $\mathcal{L}_t^\star$. Formally, given an old task $\mathcal{T}_o$, and a current parameter vector $\boldsymbol{\theta}$ we define *forgetting* as:

$$\mathcal{E}_o(\boldsymbol{\theta}) = \mathcal{L}_o(\boldsymbol{\theta}) - \mathcal{L}_o^\star \tag{1}$$

The forgetting on $\mathcal{T}_o$ after learning task $\mathcal{T}_t$ is $\mathcal{E}_o(t) := \mathcal{E}_o(\boldsymbol{\theta}_t)$ and, trivially, $\mathcal{E}_t(t) = 0$. Thus, simply, $\mathcal{E}_o(t)$ measures the reduction in performance on task $\mathcal{T}_o$ due to learning tasks $\{\mathcal{T}_{o+1}, \ldots, \mathcal{T}_t\}$. Importantly, forgetting may also be negative, in which case learning the new task has benefited the performance on the old task. Averaging over all the previous tasks we get the *average forgetting* after task $\mathcal{T}_t$: $\mathcal{E}(t) = \frac{1}{t} \sum_{o=1}^t \mathcal{E}_o(t)$, also known as *backward transfer*. The goal of continual learning algorithms is to minimise $\mathcal{E}(t)$ while learning a new task $\mathcal{T}_t$, or, equivalently, to minimise the *multi-task loss* $\mathcal{L}_{MT}(\boldsymbol{\theta}) = \sum_{o=1}^t \mathcal{L}_o(\boldsymbol{\theta})$ while only having access to the current task data $D_t$. We call $\boldsymbol{\theta}^\star$ a minimiser of the multi-task loss.

## 2 RELATED WORK

**Continual learning**  We follow De Lange et al. (2021), dividing the literature on continual learning into three families of methods: replay methods, regularization-based methods and parameter isolation methods.

*Replay methods* (Rebuffi et al., 2017; Lopez-Paz & Ranzato, 2017; Shin et al., 2017; Van de Ven et al., 2020) rely on storing a set of samples $S$ for each task, often as input or neural activity vectors. The old samples are revisited during the learning of new tasks effectively restricting the set of possible solutions to those that agree with the samples stored in memory.

*Regularisation-based methods* (Kirkpatrick et al., 2017; Zenke et al., 2017; Nguyen et al., 2017; Ebrahimi et al., 2019; Ritter et al., 2018; Guo et al., 2022) stem from a Bayesian interpretation of the continual learning problem. Given a prior over model parameters $\mathcal{P}(\boldsymbol{\theta})$ and a likelihood for the given task data $\mathcal{P}(D_t|\boldsymbol{\theta})$, learning a new task is equivalent to estimating the posterior $\mathcal{P}(\boldsymbol{\theta}|D_t)$, which is then used as prior for the next task. In practice, this different formulation results in a soft constraint on the parameters which gets incorporated in the loss function as a regulariser.

Finally, *parameter isolation methods* (Rusu et al., 2016; Yoon et al., 2017; Van Etten et al., 2018; Mallya & Lazebnik, 2018; Farajtabar et al., 2020; Wortsman et al., 2020; Saha et al., 2021; Kang et al., 2022) revolve around the idea of dynamically allocating "parts" of the network to each task $\Theta_1, \ldots, \Theta_T : \bigcup_t \Theta_t = \Theta$. Unlike the rest of the continual learning literature, we distinguish between *strict* and *general* parameter isolation. When the term *part* is understood strictly, as an index set over the parameters or the neurons, this definition comprises algorithms like PackNet (Mallya & Lazebnik, 2018), SpaceNet (Van Etten et al., 2018) or Progressive networks (Rusu et al., 2016), which employ respectively pruning, sparse training, and network expansion to isolate a small set of parameters per task. If, instead, we allow a network *part* to be any generic, non-axis aligned, subspace of the parameter space, this definition also contains algorithms like OGD (Farajtabar et al., 2020) and GPM (Saha et al., 2021) and the variants thereof (Deng et al., 2021; Lin et al., 2022), which use a more sophisticated projection mechanism to prevent forgetting.

**Geometric properties of the loss landscape**    Our results draw on existing literature studying neural networks' loss landscape. In particular, we approximate the loss function around the (local) minimum $\boldsymbol{\theta}_t$ by looking at the second order derivatives (i.e. *Hessian*) matrix. Importantly for the following discussion, Sagun et al. (2016) (also (Sagun et al., 2017)) observed that the loss landscape is mostly degenerate, especially around the optimum implying that most of the eigenvalues of the loss Hessian lie near zero. Singh et al. (2021; 2023) further established rigorously that the Hessian rank can be at most of the order square root of the number of parameters. Besides, Gur-Ari et al. (2018) analysed the evolution of the loss gradient and Hessian during training with SGD and observed that, after a short period of training, the gradient lies in the subspace spanned by the top eigenvectors of the Hessian. This subspace is shown to be approximately preserved over long periods of training, suggesting that the geometry of the loss landscape stabilises early during training.

**Catastrophic forgetting from the loss geometry viewpoint**    A few works have previously harnessed the knowledge of the loss geometry in theoretical analyses of catastrophic forgetting. Our work is similar in spirit to Yin et al. (2020), where they study regularisation-based algorithms also from a loss landscape perspective. They argue that many regularisation strategies can be interpreted as minimising a local second-order approximation to the previous tasks' loss function. Analogously, we argue that many parameter isolation strategies also inherently rely on a second-order approximation of the previous tasks' loss function, thus suggesting a substantial connection between the regularisation-based and parameter isolation approaches. Kong et al. (2023) extend the analysis in Yin et al. (2020) providing an upper-bound on forgetting for regularisation-based methods, which depends on the number of optimization iterations, the gradient variance, and the smoothness of the loss function. While Yin et al. (2020) and Kong et al. (2023) focus on the optimisation of a regularised objective, for parameter isolation, we can guarantee low levels of forgetting while being agnostic to the specific optimisation process. In this respect, parameter isolation strategies can provide stronger guarantees than regularisation-based methods.

Some empirical works have also investigated the role of geometrical properties of the found minima in the overall degree of forgetting. Mirzadeh et al. (2020b) hypothesize that forgetting correlates with the curvature of the loss function around the minimum, and thus propose to nudge the optimization algorithm towards wider minima with careful tuning of its hyper-parameters. Relying on a similar intuition, Deng et al. (2021) introduce a continual learning algorithm explicitly biased towards flatter valleys of the multitask loss. Finally, Mirzadeh et al. (2020a) study the relation between the optima of the continual and multitask settings in the light of recent work on *linear mode connectivity* in neural networks (Draxler et al., 2018; Garipov et al., 2018; Frankle et al., 2020).

Lastly, also related to this work is Bennani et al. (2020), where the authors analyse the generalisation of OGD in the NTK regime.

## 3 THE LOSS GEOMETRY PERSPECTIVE

In this section, we derive a unified framework for parameter isolation strategies based on a quadratic approximation of the loss function, and in the next section, we demonstrate that many existing algorithms are special cases of this framework. We start by the second-order Taylor expansion of forgetting $\mathcal{E}_o(\boldsymbol{\theta})$ around task $\mathcal{T}_o$ minimum $\boldsymbol{\theta}_o$:

$$\mathcal{E}_o(\boldsymbol{\theta}) = (\boldsymbol{\theta} - \boldsymbol{\theta}_o)^{\mathsf{T}}\boldsymbol{\nabla}\mathcal{L}_o(\boldsymbol{\theta}_o) + \frac{1}{2}(\boldsymbol{\theta} - \boldsymbol{\theta}_o)^{\mathsf{T}}\mathbf{H}_o^{\star}(\boldsymbol{\theta} - \boldsymbol{\theta}_o) + \boldsymbol{O}(\|\boldsymbol{\theta} - \boldsymbol{\theta}_o\|^3), \quad (2)$$

where, $\boldsymbol{\nabla}\mathcal{L}_t(\boldsymbol{\theta}) = \dfrac{\partial \mathcal{L}_t(\boldsymbol{\theta})}{\partial \boldsymbol{\theta}}$ denotes the $t$-task loss gradient vector and $\mathbf{H}_t(\boldsymbol{\theta}) = \dfrac{\partial^2 \mathcal{L}_t(\boldsymbol{\theta})}{\partial \boldsymbol{\theta} \partial \boldsymbol{\theta}^{\mathsf{T}}}$ the $t$-task loss Hessian matrix. For simplicity we shorten $\mathbf{H}_t(\boldsymbol{\theta}_t)$ to $\mathbf{H}_t^{\star}$ henceforth.

We make two assumptions: that $\boldsymbol{\theta}_o$ is a (perhaps local) minimum and that the algorithm is confined to a region of the parameter space.

**Assumption 1.** The learning algorithm terminates *in the vicinity* of a minimum for each task, i.e. $\|\boldsymbol{\nabla}\mathcal{L}_o(\boldsymbol{\theta}_o)\| = \epsilon$ and $\mathbf{H}_o^{\star} \succeq 0, \forall o \in [T]$, where $\epsilon$ is arbitrarily close to 0.

**Assumption 2.** The task solutions are close in the parameter space, i.e. there exists some (small) $\delta$ such that $\max_{i,j \in [T]} \|\boldsymbol{\theta}_i - \boldsymbol{\theta}_j\| < \delta$.

Importantly, these two assumptions are widespread in both the theoretical and empirical literature in continual learning. If they would not be satisfied, the approximation of the task loss would lose its validity. Yin et al. (2020) justify the assumptions in light of studies for overparametrised models, whose loss function appears convex around the minima. However, the validity of these two assumptions is often not checked empirically. The first assumption is mostly accurate for gradient descent and the like. Simply, the algorithm will reach a minimum if run long enough (Lee et al., 2016). The second assumption is more stringent, however, and thus it can be more easily violated. In this work, we devote a substantial part of our empirical analysis to the assessment of Assumption 2, which we consider to be of general interest for the continual learning literature.

A way to think about Assumption 2 is to consider the multi-task solution $\boldsymbol{\theta}^{\star}$, which trivially satisfies it, being a solution for all tasks. In general, in light of the work on the connectivity of individual task solutions (Mirzadeh et al., 2020a), there may exist sets of solutions $\{\boldsymbol{\theta}_1, \ldots, \boldsymbol{\theta}_T\}$ in the same region of the parameter space. Interestingly, as shown in the next section, Assumption 2 can be relaxed for some parameter isolation algorithms.

Using these assumptions we can write the average forgetting in a recursive fashion (full derivations are provided in the Appendix A.1, together with all the other proofs).

$$\mathcal{E}(t) = \frac{1}{t}\left((t-1) \cdot \mathcal{E}(t-1) + \frac{1}{2}\boldsymbol{\Delta}_t^{\mathsf{T}}(\sum_{o=1}^{t-1}\mathbf{H}_o^{\star})\boldsymbol{\Delta}_t + \boldsymbol{v}^{\mathsf{T}}\boldsymbol{\Delta}_t\right) + \mathcal{O}(\delta \cdot \epsilon) \quad (3)$$

We use $\boldsymbol{v}^{\mathsf{T}}$ to denote the vector $\sum_{o=1}^{t-2}(\boldsymbol{\theta}_{t-1} - \boldsymbol{\theta}_o)^{\mathsf{T}}\mathbf{H}_o^{\star}$. Importantly, $\boldsymbol{v}$ does not depend on the last parameter update $\boldsymbol{\Delta}_t = \boldsymbol{\theta}_t - \boldsymbol{\theta}_{t-1}$, thus this expression isolates the contribution to the average forgetting of the update due to a new task $\mathcal{T}_t$. The terms $\epsilon$ and $\delta$ are defined with the Assumptions 1-2. With this formulation, we are ready to state a first result.

**Theorem 1** (Null forgetting). *For any continual learning algorithm satisfying Assumptions 1-2 the following relationship between previous and current forgetting exists:*

$$\mathcal{E}(1), \ldots, \mathcal{E}(t-1) = 0 \implies \mathcal{E}(t) = \frac{1}{2}\boldsymbol{\Delta}_t^{\mathsf{T}}\left(\frac{1}{t} \cdot \sum_{o=1}^{t-1}\mathbf{H}_o^{\star}\right)\boldsymbol{\Delta}_t$$

Intuitively, Theorem 1 quantifies forgetting in terms of the parameter update's alignment to the *column space* of the average Hessian $\overline{\mathbf{H}}_{<t}^{\star} := \frac{1}{t-1} \cdot \sum_{o=1}^{t-1}\mathbf{H}_o^{\star}$.

An immediate consequence of this statement is that the rank of the average Hessian (ranging from 0

to $P$) gives us as a rough measure of the number of dimensions in the parameter space already "in use" by the first $t - 1$ tasks. We include in Appendix A.2 an empirical assessment of the growth of $\mathrm{rank}(\overline{\mathbf{H}}^{\star}_{<t})$ as a function of $t$.

Furthermore, unrolling Theorem 1 from $t = 1$ to $t = T$ it is straightforward to devise a constraint on the parameter update enforcing $\mathcal{E}(t) = 0$ for all $t \in [T]$. The following Theorem introduces this constraint.

**Theorem 2** (Null-forgetting constraint). *Let* $\mathcal{A} : [T] \rightarrow \Theta$ *be a continual learning algorithm satisfying Assumptions 1-2, and* $\mathcal{A}(t) := \boldsymbol{\Delta}_t$. *Then* $\mathcal{E}_\tau(t) = 0 \; \forall \; \tau < t$ *if and only if:*

$$\boldsymbol{\Delta}_t^\intercal (\sum_{o=1}^{t-1} \mathbf{H}_o^\star) \boldsymbol{\Delta}_t = 0 \tag{4}$$

Simply put, enforcing the constraint in Equation 4 guarantees zero or null forgetting as long as the assumptions are met. In the next section we argue that parameter isolation algorithms embody variations of this constraint. Before moving on to the next section we make two remarks.

First, that there's an intuitive explanation of Theorem 2: when a quadratic approximation to the loss is accurate enough, simply constraining the parameter updates to directions along which the multi-task loss landscape is flat will prevent forgetting. Importantly, the sharper the old tasks minima, the more stringent the constraint, which explains the recent interest in flat minima for continual learning (Deng et al., 2021; Mirzadeh et al., 2020a).

Second, it is easy to show that, if Assumptions 1-2 are satisfied, $\min \mathcal{E}(t) = 0$ for all $t \in [T]$. Thus, enforcing Equation 4 effectively minimises forgetting in the quadratic regime. In Appendix A.2 we consider the case where $\mathcal{E}_\tau(t) > 0$ for some $\tau < t$, and we derive a new constraint to achieve $\mathcal{E}(t) = 0$.

## 4 A UNIFIED VIEW OF PARAMETER ISOLATION METHODS

We now discuss how several existing algorithms such as OGD (Farajtabar et al., 2020), GPM (Saha et al., 2021), PackNet (Mallya & Lazebnik, 2018) and Progressive Networks (Rusu et al., 2016) can be understood as variants of our framework. For *general* parameter isolation methods the connection is less obvious, and thus we carry out a meticulous analysis of these cases. Conversely, we deem it sufficient to discuss the link to *strict* parameter isolation intuitively, using an informal proof sketch.

### 4.1 ORTHOGONAL GRADIENT DESCENT

Let $D_o = \{(\boldsymbol{x}_{1,o}, y_{1,o}), \cdots, (\boldsymbol{x}_{n_o,o}, y_{n_o,o}))\}$ be the dataset associated with task $\mathcal{T}_o$, where $y \in [C]$. For any input $\boldsymbol{x}$, we denote the network output as $\boldsymbol{f_\theta}(\boldsymbol{x}) \in \mathbb{R}^C$. The gradient of the network output with respect to the network parameters is the $P \times C$ matrix $\nabla_{\boldsymbol{\theta}} \boldsymbol{f_\theta}(\boldsymbol{x}) = \left[ \nabla_{\boldsymbol{\theta}} \boldsymbol{f}_{\boldsymbol{\theta}}^1(\boldsymbol{x}), \cdots, \nabla_{\boldsymbol{\theta}} \boldsymbol{f}_{\boldsymbol{\theta}}^C(\boldsymbol{x}) \right]$, where $P$ is the network size. The standard version of OGD imposes the following constraint on any parameter update $\boldsymbol{u}$:

$$\langle \boldsymbol{u}, \nabla_{\boldsymbol{\theta}} \boldsymbol{f}_{\boldsymbol{\theta}_o}^c(\boldsymbol{x}_{i,o}) \rangle = 0 \quad \forall c \in [C], \boldsymbol{x}_{i,o} \in D_o, o < t, \tag{5}$$

$t$ being the number of tasks solved so far. Our first result is that this constraint is equivalent to the *null-forgetting* constraint of Equation 4 whenever Assumption 1 is satisfied.

**Theorem 3.** *Let* $\mathcal{A} : [T] \rightarrow \Theta$ *be a continual learning algorithm satisfying Assumption 1. If* $\mathcal{A}(t) = \boldsymbol{\Delta}_t$ *satisfies Equation 5, then it satisfies Equation 4.*

Importantly, Assumption 2 is not necessary for this result. However, it is necessary to guarantee $\mathcal{E}(t) = 0$ given that the constraint is satisfied. As a consequence of this result, we obtain *null-forgetting* guarantees for OGD in the regime described by Assumptions 1-2.

**Corollary 1** (Null-forgetting guarantees for OGD). *Let* $\mathcal{A} : [T] \rightarrow \Theta$ *be a continual learning algorithm satisfying Assumptions 1-2. If* $\mathcal{A}(t) = \boldsymbol{\Delta}_t$ *satisfies Equation 5 for all* $t \in [T]$*, then* $\mathcal{E}(t) = 0 \, \forall t \in [T]$.

### 4.2 GRADIENT PROJECTION MEMORY

GPM stores in its memory layer-wise activation vectors for each task. Let $\boldsymbol{x}_t^l = \sigma(\boldsymbol{W}^{l-1\intercal} \boldsymbol{x}_t^{l-1})$ be the representation of $\boldsymbol{x}_t$ at the $l$-th layer of the network ($\boldsymbol{x}_t^0 = \boldsymbol{x}_t$). The GPM algorithm enforces the

following constraint on the weight matrix update $\mathbf{\Delta W}^l$:

$$\langle \mathbf{\Delta W}^l, \boldsymbol{x}_o^l \rangle = 0 \quad \forall \boldsymbol{x}_o \in D_o, o < t \tag{6}$$

We first show that the GPM constraint is equivalent to the *null-forgetting* constraint of Equation 4 when Assumption 1 is valid and if the Hessian matrix is *block-diagonal*.

**Theorem 4.** *Let* $\mathcal{A} : [T] \to \Theta$ *be a continual learning algorithm satisfying Assumption 1. If* $\mathbf{H}_o^\star$ *is* block-diagonal *for all* $o \in [T]$, *then if* $\mathcal{A}(t) = \mathbf{\Delta}_t$ *satisfies Equation 6 it also satisfies Equation 4.*

Next, we move on to show a more general result for GPM, namely that it satisfies the *null-forgetting* constraint with respect to a quadratic approximation of the multi-task loss at every point in the optimization path. As a consequence, we obtain null-forgetting guarantees for GPM *everywhere in the parameter space*, i.e. notwithstanding Assumptions 1-2.

**Theorem 5.** *Let* $\mathcal{A} : [T] \to \Theta$ *be a continual learning algorithm for a* ReLU *network and* $\mathcal{A}(t) = \mathbf{\Delta}_t = \sum_{i=1}^{S_t} \boldsymbol{\delta}_t^{(i)}$. *Moreover, let* $\{\boldsymbol{\theta}_{t-1 \to t}^{(1)}, \dots, \boldsymbol{\theta}_{t-1 \to t}^{(S_t)}\}$ *be the points on the optimization path from task* $t-1$ *to task* $t$ ($\boldsymbol{\theta}_{t-1 \to t}^{(S_t)} = \boldsymbol{\theta}_t$). *If Equation 6 is satisfied for all* $\{\boldsymbol{\delta}_t^{(s)} : s < i\}$ *then* $\forall o < t$ *it holds that:*

$$\begin{aligned} \nabla \mathcal{L}_o(\boldsymbol{\theta}_{t-1 \to t}^{(i-1)})^\mathsf{T} \boldsymbol{\delta}_t^{(i)} &= 0 \\ \boldsymbol{\delta}_t^{(i)\mathsf{T}} \, \text{block-diag}(\mathbf{H}_o(\boldsymbol{\theta}_{t-1 \to t}^{(i-1)})) \, \boldsymbol{\delta}_t^{(i)} &= 0 \end{aligned} \tag{7}$$

*where* block-diag$(\cdot)$ *denotes the layer-wise block-diagonal Hessian, i.e.,* block-diag$(\mathbf{H})_{i,j} := \mathbf{H}_{i,j} \cdot \mathbb{1}\{layer(\theta_i) = layer(\theta_j)\}$.

The constraints in Equation 7 represent the natural generalisation of the null-forgetting constraint to the case where Assumptions 1-2 are not valid. Importantly, this result only holds for *ReLU* networks, and it relies on the Hessian being block diagonal. In the Appendix C we verify empirically the extent to which this condition is met in practice.

**Corollary 2** (Null-forgetting guarantees for GPM). *Let* $\mathcal{A} : [T] \to \Theta$ *be a continual learning algorithm for a* ReLU*-network satisfying Equation 6. If* block-diag$(\mathbf{H}_o(\boldsymbol{\theta})) = \mathbf{H}_o(\boldsymbol{\theta})$ *for all* $\boldsymbol{\theta} \in \Theta$, *then* $\mathcal{E}(t) = 0 \,\forall\, t \in [T]$.

## 4.3 STRICT PARAMETER ISOLATION

As stated above, the algorithms belonging to this category partition the network by assigning different parameter sets to different tasks. The high-level intuition is that partitioning the network parameters inevitably entails orthogonality between the current task parameter update and previous tasks' updates, thus enforcing Equation 4.

More specifically, PackNet (Mallya & Lazebnik, 2018) selects a small set of task-specific parameters by pruning after training on the task in question. Alternatively, SpaceNet (Van Etten et al., 2018) directly trains a dynamically allocated subset of the network parameters for each task by enforcing sparsity. Finally Progressive Networks (Rusu et al., 2016) or other similar approaches known in the literature as *network expansion* methods, add new parameters to the network for each new task.
All these methods then freeze the parameter sets pertaining to previous tasks when learning a new task. Formally, freezing corresponds to applying the following transformation to the gradient vector: $\tilde{\nabla} \mathcal{L}_t(\boldsymbol{\theta}) \leftarrow \boldsymbol{m}_t \circ \nabla \mathcal{L}_t(\boldsymbol{\theta})$, where $\boldsymbol{m}_t$ is a binary mask $(\boldsymbol{m}_t)_i = 0$ for any $i$ in the index set of a previous task $\mathcal{I}_{<t}$ and $\circ$ denotes Hadamard-product. Similarly, the transformation applied to the Hessian matrix is: $\tilde{\boldsymbol{H}}_t(\boldsymbol{\theta}) \leftarrow \boldsymbol{M}_t \circ \boldsymbol{H}_t(\boldsymbol{\theta})$, where $\boldsymbol{M}_t = \boldsymbol{m}_t \cdot \boldsymbol{m}_t^\mathsf{T}$. Consequently, $\tilde{\nabla} \mathcal{L}_t(\boldsymbol{\theta}) \perp \tilde{\nabla} \mathcal{L}_o(\boldsymbol{\theta}_o)$ and $\tilde{\nabla} \mathcal{L}_t(\boldsymbol{\theta})^\mathsf{T} \tilde{\boldsymbol{H}}_o^\star \tilde{\nabla} \mathcal{L}_t(\boldsymbol{\theta}) = 0$. It readily follows that $\mathcal{E}_o(t) = 0$ under Assumption 2. The result carries over to the general setting described by Theorem 5 in a similar fashion, upon incorporating additional details from the algorithms.

**Enforcing sparsity** Sparsity-enforcing algorithms (Schwarz et al., 2021; Abbasi et al., 2022) constitute a special case in the parameter isolation family, as sparsity can be seen as a softer network partitioning method. As an example, Abbasi et al. (2022) enforce sparsity on the neural activations using a *k-winner* activation function and *heterogenous dropout*: in practice, with high probability, the layer-wise activation vectors between different tasks are orthogonal, i.e. $\langle \boldsymbol{x}_o^l, \boldsymbol{x}_t^l \rangle = 0$ for any pair $(o, t)$, where $\boldsymbol{x}_o \in D_o$ and $\boldsymbol{x}_t \in D_t$. Is is easy to see (cf. Equation 14 in the Appendix) that

Table 1: Approximation error, Equation 2

|  | lr | 1st order | 2nd order | Taylor | $\mathcal{E}_o(t)$ | $\|\boldsymbol{\theta}_t - \boldsymbol{\theta}_o\|$ |
|---|---|---|---|---|---|---|
| $o \in [1, t-1]$ | $1e^{-5}$ | $0.43 \pm 0.03$ | $0.26 \pm 0.02$ | $\mathbf{0.15} \pm 0.02$ | $0.30 \pm 0.03$ | $1.23 \pm 0.03$ |
|  | $1e^{-2}$ | $2.33 \pm 0.11$ | $\mathbf{1.49} \pm 0.09$ | $1.55 \pm 0.09$ | $2.26 \pm 0.10$ | $8.19 \pm 0.06$ |

this condition implies that the gradient with respect to the input weight matrix $\boldsymbol{W}^{l-1}$ is orthogonal to the previous neural activations subspace, i.e. $\nabla_{\boldsymbol{W}^{l-1}} \mathcal{L}_t \, \boldsymbol{x}_o^l = \boldsymbol{0}$, which corresponds to the GPM constraint (Equation 6). However, since the constraint is only met in probability, this algorithm can only offer probabilistic guarantees.

## 5 EXPERIMENTS AND RESULTS

In this section, we present experimental results validation our theoretical findings. First, we assess the validity of Assumption 2, and evaluate the accuracy of a quadratic approximation of the loss. Then we validate two claims hidden in our findings: (1) that that OGD and GPM are essentially equivalent to enforcing the *null-forgetting* constraint when Assumptions 1-2 are met and (2) that our results are agnostic of the architecture or the number of tasks.

**Experimental setup.** For all experiments, we report the mean and standard deviation over 5 runs with different seeds. For brevity, the detailed instructions for reproducing the results are in Appendix B. To facilitate the results' interpretation, we report forgetting in terms of accuracy. Moreover, we always measure forgetting on the test data.

**Datasets.** We perform our experiments on standard continual learning benchmarks. Namely, Rotated MNIST, Split CIFAR-10 and Split CIFAR-100. Rotated MNIST consists of a sequence of 5 tasks, each obtained by applying a fixed image rotation to the MNIST dataset. To avoid an additional source of randomness in the experiments, we fix the set of rotations. The split challenges are obtained by partitioning the dataset's classes into separate tasks. We split CIFAR-10 and CIFAR-100 into 5 and 20 tasks respectively.

**Networks.** For the experiments on Rotated MNIST we are able to compute the full Hessian matrix and its eigen-decomposition. We use a small feed-forward neural network with 3 hidden layers of width 50 and we downscale the input image to $14 \times 14$. In addition, we perform experiments on a convolutional neural network, a ResNet 18 (He et al., 2016) and a Vision Transformer, ViT, (Dosovitskiy et al., 2020) (see Section 5.3). For these larger networks we use deflated power iteration (Yao et al., 2018b) for computing the top-$k$ eigenvectors of the Hessian.

### 5.1 EMPIRICAL ASSESSMENT OF THE VALIDITY OF A LOCAL QUADRATIC APPROXIMATION OF THE LOSS

Many theoretical and empirical works in continual learning rely on a quadratic approximation of the loss function around the old tasks optima. Yin et al. (2020) show that this is the case for the regularisation family, and in this work we have argued that it also holds for some methods in the parameter isolation family. In this section we question this assumption, empirically testing its limits.

First, we measure the accuracy of the second order Taylor approximation (Equation 2) to predict $\mathcal{E}_o(t)$ for $t = 1, \ldots, T$. In Table 1 we report the prediction error $|\hat{\mathcal{E}}_o(t) - \mathcal{E}_o(t)|$ for SGD using a low and high learning rate on the Rotated-MNIST challenge.
In general, the approximation is more accurate for low learning rates, which limit the update norm (cf. $\|\boldsymbol{\theta}_t - \boldsymbol{\theta}_o\|$ in Table 1). Moreover, in Table 1 we inspect the contribution of the first- and second-order terms to the prediction. Remarkably, the second-order term achieves the lowest error on average in the high learning rate case. We conjecture that this effect is due to the faster rate of convergence of the second-order derivatives of the loss while training (Gur-Ari et al., 2018), which leads to more stable estimates across an extended region of the landscape, as compared to the high variance of the gradient vectors.
Unfortunately, we cannot evaluate the approximation accuracy in this way for larger networks since

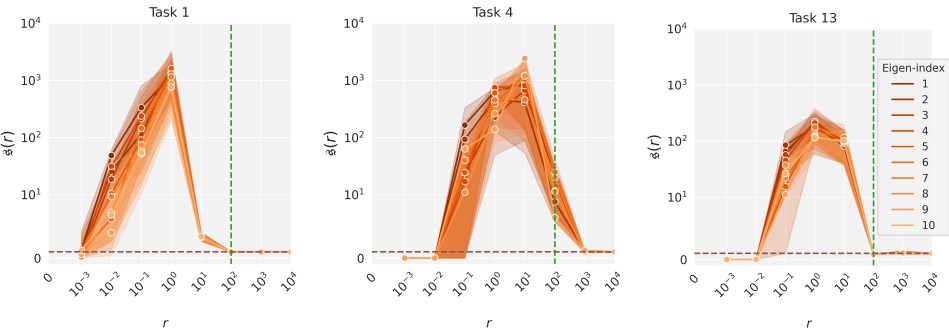

Figure 2: Perturbation score $\mathfrak{s}(r)$ varying the radius $r$ on a logarithmic scale in the range $[10^{-3}, 10^4]$ on different tasks from different challenges. The challenges are solved from left to right respectively with an MLP, a ViT and a ResNet. The confidence intervals display the uncertainty over 5 seeds. The dashed green line highlights the value of $r$ for which the score converges to 1 (marked by the dashed brown line).

computing the Hessian matrix is expensive. We thus devise another experiment which can be carried out on larger networks as well.

**Perturbation analysis**   With this experiment we measure the extent to which Assumption 2 can effectively be violated. We perturb a task solution $\boldsymbol{\theta}_t$ along the Hessian principle eigenvectors $\boldsymbol{v}_i$ and along random directions $\boldsymbol{\mu} \sim \mathcal{N}(0, \boldsymbol{I}_P)$, controlling the radius of the perturbation through a parameter $r$. We then compare the effect on the loss function through the following score:

$$\mathfrak{s}(r) = \frac{|\mathcal{L}_t(\boldsymbol{\theta}_t + r \cdot \boldsymbol{v}_i) - \mathcal{L}_t(\boldsymbol{\theta}_t)|}{\mathbb{E}_{\boldsymbol{\mu}}\left[|\mathcal{L}_t(\boldsymbol{\theta}_t + r \cdot \boldsymbol{\mu}') - \mathcal{L}_t(\boldsymbol{\theta}_t)|\right]}$$

where $\boldsymbol{\mu}' := \frac{\boldsymbol{\mu}}{||\boldsymbol{\mu}||}$. In Figure 2 we plot $\mathfrak{s}(r)$ for several values of $r$ on some tasks of the three challenges we consider. More plots can be found in Appendix C. Interestingly, the same trend can be observed across tasks and across architectures. These observations confirm that in a ball around the optimum the loss Hessian principal eigenvectors represent the directions where the loss landscape is steeper, and, conversely, that outside of this ball the second order approximation to the loss breaks down. These empirical results suggest that local quadratic approximations are still accurate reasonably far away from their center, however they also show their tangible limits. Importantly, across tasks and random seeds the size of the ball $B$ appears to be stable, while increasing proportionally to the network size. Investigating this effect further is outside of the scope of this analysis, however the use of tools such as the *perturbation analysis* may help us gain insights into still unknown geometric properties of the loss landscapes.

## 5.2   OGD and GPM satisfy the null-forgetting constraint

We augment SGD with a projection step, enforcing orthogonality to the first $k$ eigenvectors of the Hessian of each task. We refer to the augmented version of SGD as SGD$^\dagger$. In formula:

$$\text{SGD} : \boldsymbol{\theta}' \leftarrow \boldsymbol{\theta} - \eta\boldsymbol{g} \quad \text{and} \quad \text{SGD}^\dagger : \boldsymbol{\theta}' \leftarrow \boldsymbol{\theta} - \eta(\boldsymbol{I} - \boldsymbol{M}\boldsymbol{M}^\intercal)\boldsymbol{g}, \qquad (8)$$

where $\boldsymbol{g}$ represents the batch gradient vector of the current task $\mathcal{T}_t$, and $\boldsymbol{M}$ is a basis for $\text{span}(\{\boldsymbol{v}_1^o, \ldots, \boldsymbol{v}_k^o \mid o \in [1, t-1], \boldsymbol{v}_i^o := i\text{-th eigenvector of } \mathbf{H}_o^\star\})$. We follow the choice by Saha et al. (2021) to determine $k$ dynamically as a function of the spectral energy of $\mathbf{H}_o^\star$, modifying OGD accordingly. Intuitively, we consider the first $(1 - \epsilon)$ fraction of the Hessian spectral energy. We refer the reader to Appendix B for further details. Importantly, since SGD$^\dagger$ operates with the Hessian matrix at the task optimum, it will not work when Assumption 2 is violated.

We evaluate GPM, OGD and SGD$^\dagger$ on the Rotated-MNIST challenge against the SGD baseline. In order to compare the case in which Assumption 2 holds against the case in which it doesn't we train the models twice, changing the learning rate from 0.01 to $10^{-5}$, while keeping the number of epochs fixed. Additionally, in the Appendix C we also evaluate Equation 4 for OGD and GPM.

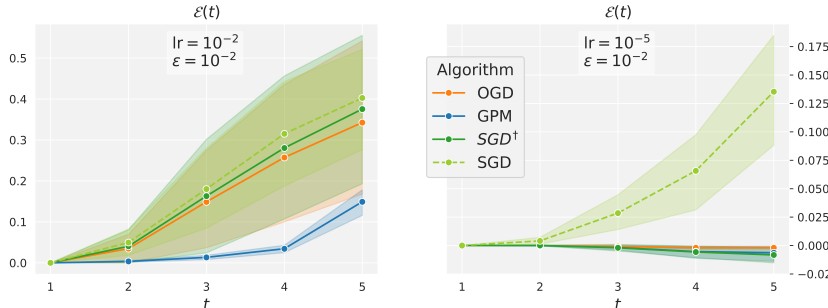

Figure 3: Comparison of all the methods in the high (left) and low (right) learning rate (lr) setting. On the $y$ axes, the accuracy average forgetting. Notice that the levels of forgetting are overall lower in the low learning rate setting.

| Network | Algorithm | CIFAR-10 ($\mathcal{E}(5)$) | CIFAR-100 ($\mathcal{E}(20)$) |
|---|---|---|---|
| | SGD | $0.029 \pm 0.005$ | $0.29 \pm 0.01$ |
| ResNet | SGD$^\dagger$-10 | $0.026 \pm 0.004$ | $0.22 \pm 0.008$ |
| | SGD$^\dagger$-20 | $\mathbf{0.021} \pm 0.007$ | $\mathbf{0.16} \pm 0.01$ |
| | SGD | $0.026 \pm 0.044$ | $0.25 \pm 0.01$ |
| ViT | SGD$^\dagger$-10 | $0.006 \pm 0.004$ | $0.16 \pm 0.02$ |
| | SGD$^\dagger$-20 | $\mathbf{0.004} \pm 0.004$ | $\mathbf{0.11} \pm 0.02$ |

Table 2: Average forgetting $\mathcal{E}(T)$ measured in terms of accuracy of SGD and SGD$^\dagger$ with varying values of $k$ on Split CIFAR-10 ($T = 5$) and Split CIFAR-100 ($T = 20$). The relatively high variance is partly due to the randomness inherent in the iterative eigen-decomposition process.

We plot the results in Figure 3. When the learning rate is high, OGD and SGD$^\dagger$, which rely on Assumption 2, perform similarly to SGD. In line with our theoretical findings, we observe low forgetting for GPM in both regimes. When the learning rate is low the three methods are equivalent. These results confirm our theoretical findings that OGD and GPM can be explained by the constraint in Equation 4.

## 5.3 RESULTS FOR OTHER ARCHITECTURES

The power of a theory is commensurate with its predictive scope. In this last section we want to demonstrate that our theoretical findings are not bound to any specific architecture, or any fixed number of tasks. In order to do so we would like to use SGD$^\dagger$ to train a convolutional or transformer network. However, enforcing Equation 4 is too computationally expensive on any network of decent size. Therefore we instead enforce an approximation of Equation 4 using only the first 10 or 20 eigenvectors of each task Hessian matrix: we refer to these approximation respectively as SGD$^\dagger$-10 and SGD$^\dagger$-20.

Moreover, in order to satisfy Assumption 2 we again train with a low learning rate (0.001). Overall, this training setup is suboptimal in many respects (performance, computation, memory) and it is designed specifically to validate the theoretical findings. Examples of efficient algorithms implementing the null-forgetting constraint are OGD, GPM and the other algorithms discussed, and competing with existing algorithms is not a goal of this study.

The results are summarised in Table 2, and an extended version can be found in Appendix C. We observe a reduction in final forgetting when using SGD$^\dagger$ during training. Importantly, this reduction is visible regardless of the architecture or the number of tasks (5 or 20) to be solved, suggesting that the null-forgetting constraint is valid for different settings.

SGD$^\dagger$-10 or -20 implement a significantly low-rank approximation of the Hessian matrix, given that its dimension is $P \times P$ and $P$ can be in the order of $10^6$. The fact that the effect is stronger when using 20 instead of 10 dimensions confirms that the quality of the Hessian approximation plays a role in the effectiveness of the constraint.

## 6 CONCLUSION

We propose a unified theoretical framework for parameter isolation strategies in continual learning, based on a quadratic loss approximation perspective. We show that many existing parameter isolation algorithms are special cases of our framework, thereby providing theoretical grounding for them. Furthermore, we use this framework to establish guarantees on catastrophic forgetting for two such algorithms. Our results also highlight a non trivial connection of parameter isolation and regularisation-based algorithms, which we invite future research to explore.

### ACKNOWLEDGEMENTS

Sidak Pal Singh would like to acknowledge the financial support from Max Planck ETH Center for Learning Systems.

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

# Appendix

This document contains the materials supporting and extending the discussion in the main text. It is organised as follows:

- **Appendix A** collects the proofs to all the Theorems and Corollaries of the main text. Additionally, we elaborate further observations derived from our theoretical framework.
- **Appendix B** provides all the details of our experimental setup.
- **Appendix C** gathers additional results supporting the experiments described in the main text.

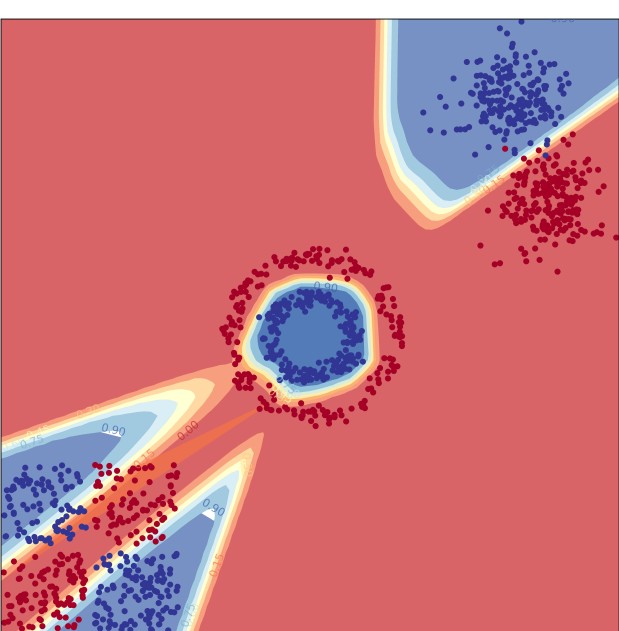

Figure 4: The multitask solution for the toy challenge in Figure 1. The same network has been used. This demonstrates that a multitask solution exists, despite standard optimization algorithms are not able to find it.

# A PROOFS AND FURTHER DISCUSSION

## A.0.1 NOTATION AND FORMALISM

First, we shortly recap the notation and formalism introduced in our discussion.

We consider a sequence of supervised learning tasks $\mathcal{T}_1, \ldots, \mathcal{T}_T$, each associated with a dataset $D_t = \{(x_1, y_1), \ldots, (x_{n_t}, y_{n_t})\}$, where $(x, y) \in \mathcal{X} \times \mathcal{Y}_t$. We describe a neural network by its set of parameters $\Theta_t \subseteq \mathbb{R}^P$, $P$ denoting the total number of parameters in the network. We tipically use the index $t$ to refer to the task currently being learned and $o$ to a general previous task.

We use bold to indicate a vector or a matrix, e.g. $\mathbf{0}$ is the null vector. $\boldsymbol{\theta}$ refers to a general value taken by the parameters, and $\boldsymbol{\theta}_t$ is the solution found on task $\mathcal{T}_t$. Moreover, we write $\boldsymbol{\Delta}_t := \boldsymbol{\theta}_t - \boldsymbol{\theta}_{t-1}$, i.e. the update to the parameters following task $\mathcal{T}_t$. Simply, $\boldsymbol{\Delta}_0 = \mathbf{0}$, $\boldsymbol{\theta}_0$ being the initialisation point.

We use $l_t(x, y, \boldsymbol{\theta})$ to denote the $t$-th task loss function, and write the average loss over the dataset as $\mathcal{L}_t(\boldsymbol{\theta})$, i.e. $\mathcal{L}_t(\boldsymbol{\theta}) = \frac{1}{n_t} \sum_{i=1}^{n_t} l_t(x_i, y_i, \boldsymbol{\theta})$. We shorten the loss measured at the end of training $\mathcal{L}_t(\boldsymbol{\theta}_t)$ to $\mathcal{L}_t^\star$.

The vector of first-order derivatives or gradient of the loss with respect to the parameters $\boldsymbol{\theta}$ is $\nabla \mathcal{L}_t(\boldsymbol{\theta}) = \frac{\partial \mathcal{L}_t(\boldsymbol{\theta})}{\partial \boldsymbol{\theta}}$ and the matrix of second-order-derivativatives or Hessian of the loss with respect to the parameters $\boldsymbol{\theta}$ is $\mathbf{H}_t(\boldsymbol{\theta}) = \frac{\partial^2 \mathcal{L}_t(\boldsymbol{\theta})}{\partial \boldsymbol{\theta}^2}$. In the interest of space, we shorten the Hessian evaluated at the end of training $\mathbf{H}_t(\boldsymbol{\theta}_t)$ to $\mathbf{H}_t^\star$.

In our discussion we use w.l.o.g. $\Theta_t \subseteq \Theta$. We will now justify this choice. Consider the case in which the network is somewhat different for each task (e.g. in the case of Progressive Networks (Rusu et al., 2016) a new component is added to the network for each task). We then write $\Theta_i \neq \Theta_j$ for any $i, j \in [T]$. Define $\Theta = \bigcup_{t=1}^T \Theta_t$ to be the *final parameter space*. Accordingly, each task update $\boldsymbol{\Delta}_t$ belongs to a subspace of the final parameter space, i.e. $\boldsymbol{\theta}_t \in \Theta_t \subseteq \Theta$. Our analysis applies to a general $\Theta$, thus carrying over to any choice of $\Theta_t$. Intuitively, using new parameters for each task introduces orthogonality between the parameters (and correspondingly their update vectors) over task-specific dimensions in $\Theta$. In order to be comprehensive, we use a general $\Theta$ in our analysis.

## A.1 OMITTED PROOFS

### A.1.1 APPROXIMATIONS OF FORGETTING

In the following, we show the steps leading to Equation 3.

We start from the Taylor expansion (Equation 2) of the loss around $\boldsymbol{\theta}_t$:

$$\mathcal{E}_o(\boldsymbol{\theta}_t) = (\boldsymbol{\theta}_t - \boldsymbol{\theta}_o)^\mathsf{T} \nabla \mathcal{L}_o(\boldsymbol{\theta}_o) + \frac{1}{2}(\boldsymbol{\theta}_t - \boldsymbol{\theta}_o)^\mathsf{T} \mathbf{H}_o^\star (\boldsymbol{\theta}_t - \boldsymbol{\theta}_o) + \boldsymbol{O}(\|\boldsymbol{\theta}_t - \boldsymbol{\theta}_o\|^3)$$

Using Assumptions 1-2 we write:

$$\begin{aligned}
\mathcal{E}_o(t) &= \frac{1}{2}(\boldsymbol{\theta}_t - \boldsymbol{\theta}_o)^\mathsf{T} \mathbf{H}_o^\star (\boldsymbol{\theta}_t - \boldsymbol{\theta}_o) + \mathcal{O}(\delta \cdot \epsilon) \\
&= \frac{1}{2}\left(\sum_{\tau=o+1}^{t} \boldsymbol{\Delta}_\tau\right)^\mathsf{T} \mathbf{H}_o^\star \left(\sum_{\tau=o+1}^{t} \boldsymbol{\Delta}_\tau\right) + \mathcal{O}(\delta \cdot \epsilon) \\
&= \underbrace{\frac{1}{2}\left(\sum_{\tau=o+1}^{t-1} \boldsymbol{\Delta}_\tau\right)^\mathsf{T} \mathbf{H}_o^\star \left(\sum_{\tau=o+1}^{t-1} \boldsymbol{\Delta}_\tau\right)}_{\mathcal{E}_o(t-1)} + \frac{1}{2}\boldsymbol{\Delta}_t^\mathsf{T} \mathbf{H}_o^\star \boldsymbol{\Delta}_t + \\
&\quad + \frac{1}{2}\left(\sum_{\tau=o+1}^{t-1} \boldsymbol{\Delta}_\tau\right)^\mathsf{T} \mathbf{H}_o^\star \boldsymbol{\Delta}_t + \frac{1}{2}\boldsymbol{\Delta}_t^\mathsf{T} \mathbf{H}_o^\star \left(\sum_{\tau=o+1}^{t-1} \boldsymbol{\Delta}_\tau\right) + \mathcal{O}(\delta \cdot \epsilon) \\
&= \mathcal{E}_o(t-1) + \frac{1}{2}\boldsymbol{\Delta}_t^\mathsf{T} \mathbf{H}_o^\star \boldsymbol{\Delta}_t + \left(\sum_{\tau=o+1}^{t-1} \boldsymbol{\Delta}_\tau\right)^\mathsf{T} \mathbf{H}_o^\star \boldsymbol{\Delta}_t + \mathcal{O}(\delta \cdot \epsilon)
\end{aligned}$$

Based on this formulation, we characterize the average forgetting as follows:

$$
\begin{aligned}
\mathcal{E}(t) &= \frac{1}{t}\sum_{o=1}^{t}\mathcal{E}_o(t) = \frac{1}{t}\underbrace{\mathcal{E}_t(t)}_{=0} + \frac{1}{t}\sum_{o=1}^{t-1}\mathcal{E}_o(t) \\
&= \frac{1}{t}\sum_{o=1}^{t-1}\left[\mathcal{E}_o(t-1) + \frac{1}{2}\boldsymbol{\Delta}_t^\mathsf{T}\mathbf{H}_o^\star\boldsymbol{\Delta}_t + \left(\sum_{\tau=o+1}^{t-1}\boldsymbol{\Delta}_\tau\right)^\mathsf{T}\mathbf{H}_o^\star\boldsymbol{\Delta}_t + \mathcal{O}(\delta\cdot\epsilon)\right] \\
&= \frac{1}{t}\sum_{o=1}^{t-1}[\mathcal{E}_o(t-1)] + \frac{1}{t}\sum_{o=1}^{t-1}\left[\frac{1}{2}\boldsymbol{\Delta}_t^\mathsf{T}\mathbf{H}_o^\star\boldsymbol{\Delta}_t\right] + \frac{1}{t}\sum_{o=1}^{t-1}\left[\left(\sum_{\tau=o+1}^{t-1}\boldsymbol{\Delta}_\tau\right)^\mathsf{T}\mathbf{H}_o^\star\boldsymbol{\Delta}_t\right] + \mathcal{O}(\delta\cdot\epsilon) \\
&= \frac{t-1}{t}\cdot\mathcal{E}(t-1) + \frac{1}{2t}\boldsymbol{\Delta}_t^\mathsf{T}\left[\sum_{o=1}^{t-1}\mathbf{H}_o^\star\right]\boldsymbol{\Delta}_t + \frac{1}{t}\left[\underbrace{\sum_{o=1}^{t-1}(\boldsymbol{\theta}_{t-1}-\boldsymbol{\theta}_o)^\mathsf{T}\mathbf{H}_o^\star}_{\boldsymbol{v}^\mathsf{T}}\right]\boldsymbol{\Delta}_t + \mathcal{O}(\delta\cdot\epsilon) \\
&= \frac{1}{t}\left((t-1)\cdot\mathcal{E}(t-1) + \frac{1}{2}\boldsymbol{\Delta}_t^\mathsf{T}(\sum_{o=1}^{t-1}\mathbf{H}_o^\star)\boldsymbol{\Delta}_t + \boldsymbol{v}^\mathsf{T}\boldsymbol{\Delta}_t\right) + \mathcal{O}(\delta\cdot\epsilon)
\end{aligned}
$$

### A.1.2 THEOREM 1

We want to prove the following statement.

*Theorem 1* (Null forgetting). For any continual learning algorithm satisfying Assumptions 1-2 the following relationship between previous and current forgetting exists:

$$
\mathcal{E}(1),\ldots,\mathcal{E}(t-1) = 0 \implies \mathcal{E}(t) = \frac{1}{2}\boldsymbol{\Delta}_t^\mathsf{T}\left(\frac{1}{t}\cdot\sum_{o=1}^{t-1}\mathbf{H}_o^\star\right)\boldsymbol{\Delta}_t
$$

**Proof.** We prove the above statement by induction. We prove the base case for $t = 1$ and $t = 2$, since some terms trivially cancel out for $t = 1$.

*Base case 1*: $\mathcal{E}(1) = 0 \implies \mathcal{E}(2) = \frac{1}{2}\boldsymbol{\Delta}_2^\mathsf{T}H_1^\star\boldsymbol{\Delta}_2$. Notice that by definition, $\mathcal{E}(1) = \mathcal{E}_1(1) = 0$.

Using Equation 3 we can write $\mathcal{E}(2)$:

$$
\begin{aligned}
\mathcal{E}(2) &= \mathcal{E}(1) + \frac{1}{2}\boldsymbol{\Delta}_2^\mathsf{T}(H_1^\star)\boldsymbol{\Delta}_2 + (\boldsymbol{\theta}_1 - \boldsymbol{\theta}_1)^\mathsf{T}H_1^\star\boldsymbol{\Delta}_2 \\
&= 0 + \frac{1}{2}\boldsymbol{\Delta}_2^\mathsf{T}H_1^\star\boldsymbol{\Delta}_2 + 0
\end{aligned}
$$

*Base case 2*: $\mathcal{E}(1) = 0, \mathcal{E}(2) = 0 \implies \mathcal{E}(3) = \frac{1}{2}\boldsymbol{\Delta}_3^\mathsf{T}(\frac{1}{3}H_1^\star + \frac{1}{3}H_2^\star)\boldsymbol{\Delta}_3$.
Using the last result from case 1, we have that :

$$
\mathcal{E}(2) = \frac{1}{2}\boldsymbol{\Delta}_2^\mathsf{T}H_1^\star\boldsymbol{\Delta}_2 = 0
$$

The latter equation implies that $\boldsymbol{\Delta}_2^\mathsf{T}H_1^1 = \mathbf{0}$. Plugging it into the value of $\mathcal{E}(3)$ given by Equation 3:

$$
\begin{aligned}
\mathcal{E}(3) &= \frac{1}{3}\left(2\cdot\mathcal{E}(2) + \frac{1}{2}\boldsymbol{\Delta}_3^\mathsf{T}(H_2^\star + H_1^\star)\boldsymbol{\Delta}_3 + \sum_{t=1}^{2}(\boldsymbol{\theta}_2 - \boldsymbol{\theta}_t)^\mathsf{T}H_t^\star\boldsymbol{\Delta}_3\right) \\
&= 0 + \frac{1}{2}\boldsymbol{\Delta}_3^\mathsf{T}(\frac{1}{3}H_2^\star + \frac{1}{3}H_1^\star)\boldsymbol{\Delta}_3 + \frac{1}{3}\underbrace{\boldsymbol{\Delta}_2^\mathsf{T}H_1^1}_{=\mathbf{0}}\boldsymbol{\Delta}_3 + \frac{1}{3}\underbrace{(\boldsymbol{\theta}_2 - \boldsymbol{\theta}_2)}_{=\mathbf{0}}^\mathsf{T}H_t^\star\boldsymbol{\Delta}_3
\end{aligned}
$$

*Induction step: if $\mathcal{E}(\tau) = 0\ \forall \tau < t$ then $\mathcal{E}(t) \geq 0$.*

We start by writing out $\mathcal{E}(t)$.

$$\mathcal{E}(t) = \frac{1}{t}\left((t-1)\cdot\mathcal{E}(t-1) + \frac{1}{2}\boldsymbol{\Delta}_t^{\mathsf{T}}\left(\sum_{\tau=1}^{t-1}\boldsymbol{H}_\tau^\star\right)\boldsymbol{\Delta}_t + \sum_{\tau=1}^{t-1}(\boldsymbol{\theta}_{t-1} - \boldsymbol{\theta}_\tau)^{\mathsf{T}}\boldsymbol{H}_\tau^\star\boldsymbol{\Delta}_t\right)$$

$$= 0 + \frac{1}{2}\boldsymbol{\Delta}_t^{\mathsf{T}}\left(\sum_{\tau=1}^{t-1}\boldsymbol{H}_\tau^\star\right)\boldsymbol{\Delta}_t + \sum_{\tau=1}^{t-1}(\boldsymbol{\Delta}_{\tau+1} + \cdots + \boldsymbol{\Delta}_{t-1})^{\mathsf{T}}\boldsymbol{H}_\tau^\star\boldsymbol{\Delta}_t$$

The induction step is accomplished if $\sum_{\tau=1}^{t-1}(\boldsymbol{\Delta}_{\tau+1} + \cdots + \boldsymbol{\Delta}_{t-1})^{\mathsf{T}}\boldsymbol{H}_\tau^\star = \sum_{\tau=1}^{t-2}(\boldsymbol{\Delta}_{\tau+1} + \cdots + \boldsymbol{\Delta}_{t-1})^{\mathsf{T}}\boldsymbol{H}_\tau^\star = \boldsymbol{0}$. Consider now the case in which $\sum_{\tau=1}^{t-1}(\boldsymbol{\Delta}_{\tau+1} + \cdots + \boldsymbol{\Delta}_{t-1})^{\mathsf{T}}\boldsymbol{H}_\tau^\star \neq \boldsymbol{0}$. It follows that:

$$\sum_{\tau=1}^{t-2}(\boldsymbol{\Delta}_{\tau+1} + \cdots + \boldsymbol{\Delta}_{t-1})^{\mathsf{T}}\boldsymbol{H}_\tau^\star \neq \boldsymbol{0}$$

$$\sum_{\tau=1}^{t-2}\boldsymbol{\Delta}_{t-1}^{\mathsf{T}}\boldsymbol{H}_\tau^\star + \sum_{\tau=1}^{t-2}(\boldsymbol{\Delta}_{\tau+1} + \cdots + \boldsymbol{\Delta}_{t-2})^{\mathsf{T}}\boldsymbol{H}_\tau^\star \neq \boldsymbol{0}$$

Multilying by $\boldsymbol{\Delta}_{t-1}$ on the right we get:

$$\sum_{\tau=1}^{t-2}\boldsymbol{\Delta}_{t-1}^{\mathsf{T}}\boldsymbol{H}_\tau^\tau\boldsymbol{\Delta}_{t-1} + \sum_{\tau=1}^{t-3}(\boldsymbol{\Delta}_{\tau+1} + \cdots + \boldsymbol{\Delta}_{t-2})^{\mathsf{T}}\boldsymbol{H}_\tau^\tau\boldsymbol{\Delta}_{t-1} \neq \boldsymbol{0}^{\mathsf{T}}\boldsymbol{\Delta}_{t-1}$$

We compare this result with the value of $\mathcal{E}(t-1)$ given by Equation 3:

$$\mathcal{E}(t-1) = \frac{1}{t-1}\left((t-2)\cdot\mathcal{E}(t-2) + \frac{1}{2}\boldsymbol{\Delta}_{t-1}^{\mathsf{T}}\left(\sum_{o=1}^{t-2}\boldsymbol{H}_o^\star\right)\boldsymbol{\Delta}_{t-1} + \sum_{\tau=1}^{t-2}(\boldsymbol{\Delta}_{\tau+1} + \cdots + \boldsymbol{\Delta}_{t-1})^{\mathsf{T}}\boldsymbol{H}_\tau^\star\boldsymbol{\Delta}_{t-1}\right)$$

$$= \frac{1}{t-1}\left(0 + \frac{1}{2}\boldsymbol{\Delta}_{t-1}^{\mathsf{T}}\left(\sum_{\tau=1}^{t-2}\boldsymbol{H}_\tau^\star\right)\boldsymbol{\Delta}_{t-1} + \sum_{\tau=1}^{t-2}(\boldsymbol{\Delta}_{\tau+1} + \cdots + \boldsymbol{\Delta}_{t-1})^{\mathsf{T}}\boldsymbol{H}_\tau^\star\boldsymbol{\Delta}_{t-1}\right) \neq 0$$

We arrived at a contradiction, which proves the induction step and concludes the claim's proof.

### A.1.3 THEOREM 2

In order to arrive at the statement of Theorem 2 simply notice that, by definition, $\mathcal{E}(1) = 0$ for any continual learning algorithm, and apply repeatedly Theorem 1. We briefly go through all the steps.

By $\mathcal{E}(1) = 0$ and Theorem 1, $\mathcal{E}(2) = \boldsymbol{\Delta}_2^{\mathsf{T}}(\sum_{o=1}^{1}\boldsymbol{H}_o^\star)\boldsymbol{\Delta}_2$. By assumption 1-2, all Hessian matrices are positive semi-definite and, accordingly, $\mathcal{E}(2) \geq 0$. It follows that $\mathcal{A}$ is forgetting-optimal if and only if $\mathcal{E}(2) = 0$. Recursively applying this argument, if $\mathcal{A}$ is forgetting-optimal then it holds:

$$\mathcal{E}(2) = \boldsymbol{\Delta}_2^{\mathsf{T}}\left(\sum_{o=1}^{1}\boldsymbol{H}_o^\star\right)\boldsymbol{\Delta}_2 = 0$$

$$\cdots$$

$$\mathcal{E}(T) = \boldsymbol{\Delta}_T^{\mathsf{T}}\left(\sum_{o=1}^{T-1}\boldsymbol{H}_o^\star\right)\boldsymbol{\Delta}_T = 0$$

as stated in Theorem 2.

### A.1.4 THEOREM 3

We wish to prove the following statement regarding Orthogonal Gradient Descent (OGD) Farajtabar et al. (2020).

*Theorem 3.* Let $\mathcal{A} : [T] \to \Theta$ be a continual learning algorithm satisfying Assumption 1. If $\mathcal{A}(t) = \boldsymbol{\Delta}_t$ satisfies Equation 5, then it satisfies Equation 4.

We start by recalling the OGD algorithm. Let $D_o = \{(\boldsymbol{x}_1, y_1), \cdots, (\boldsymbol{x}_{n_o}, y_{n_o}))\}$ be the dataset associated with task $\mathcal{T}_o$ and $\boldsymbol{f_\theta}(\boldsymbol{x}) \in \mathbb{R}^K$ be the network output corresponding to the input $\boldsymbol{x}$. The gradient of the network output with respect to the network parameters is the $P \times K$ matrix:

$$\nabla_{\boldsymbol{\theta}} \boldsymbol{f_\theta}(\boldsymbol{x}) = \left[ \nabla_{\boldsymbol{\theta}} \boldsymbol{f}_{\boldsymbol{\theta}}^1(\boldsymbol{x}), \cdots, \nabla_{\boldsymbol{\theta}} \boldsymbol{f}_{\boldsymbol{\theta}}^K(\boldsymbol{x}) \right]$$

The standard version of OGD imposes the following constraint on any parameter update $\boldsymbol{u}$:

$$\langle \boldsymbol{u}, \nabla_{\boldsymbol{\theta}} \boldsymbol{f}_{\boldsymbol{\theta}_o}^k(\boldsymbol{x}_i) \rangle = 0$$

for all $k \in [1, K]$, $\boldsymbol{x}_i \in D_o$ and $o \le t$, $t$ being the number of tasks solved so far. If SGD is used, for example, the update $\boldsymbol{u}$ is the gradient of the new task loss for a data batch. Notice that the old task gradients are evaluated at the minima $\boldsymbol{\theta}_o$.

Proof Recall the definition of the average loss:

$$\mathcal{L}_t(\boldsymbol{\theta}) = \frac{1}{n_t} \sum_{i=0}^{n_t} l_t(\boldsymbol{x}_{i,t}, y_{i,t}, \boldsymbol{\theta})$$

The Hessian matrix of the loss $\boldsymbol{H}_t(\boldsymbol{\theta})$ can be decomposed as a sum of two other matrices (Schraudolph, 2002): the *outer-product* Hessian and the *functional* Hessian.

$$\boldsymbol{H}_t(\boldsymbol{\theta}) = \frac{1}{n_t} \sum_{i=1}^{n_t} \nabla_{\boldsymbol{\theta}} \boldsymbol{f_\theta}(\boldsymbol{x}_i) \left[ \nabla_{\boldsymbol{f}}^2 \ell_i \right] \nabla_{\boldsymbol{\theta}} \boldsymbol{f_\theta} \left( \boldsymbol{x}_i \right)^\intercal + \frac{1}{n_t} \sum_{i=1}^{n_t} \sum_{k=1}^{K} [\nabla_{\boldsymbol{f}} \ell_i]_k \, \nabla_{\boldsymbol{\theta}}^2 \boldsymbol{f}_{\boldsymbol{\theta}}^k \left( \boldsymbol{x}_i \right), \qquad (9)$$

where $\ell_i = l(\boldsymbol{x_i}, y_i)$ and $\nabla_{\boldsymbol{f}}^2 \ell_i$ is the $K \times K$ matrix of second order derivatives of the loss $\ell_i$ with respect to the network output $\boldsymbol{f_\theta}(\boldsymbol{x_i})$. At the optimum $\boldsymbol{\theta}_t$ (Assumption 1) the contribution of the functional Hessian is negligible (Singh et al., 2021). We rewrite the goal of the proof using these two facts:

$$\boldsymbol{\Delta}_t^\intercal \boldsymbol{H}_o^\star \boldsymbol{\Delta}_t = 0$$

$$\boldsymbol{\Delta}_t^\intercal \left( \frac{1}{n_o} \sum_{i=1}^{n_o} \nabla_{\boldsymbol{\theta}} \boldsymbol{f_\theta}(\boldsymbol{x}_i) \left[ \nabla_{\boldsymbol{f}}^2 \ell_i \right] \nabla_{\boldsymbol{\theta}} \boldsymbol{f_\theta} \left( \boldsymbol{x}_i \right)^\intercal \right) \boldsymbol{\Delta}_t = 0 \qquad (10)$$

Farajtabar et al. (2020) apply the OGD constraint (Equation 5) to the batch gradient vector $\boldsymbol{g}_B = \nabla \mathcal{L}_t^B$. Following this choice $\boldsymbol{\Delta}_t = \sum_{s=1}^{S_t} -\eta \boldsymbol{g}_s$, where $\eta$ is the learning rate. Clearly, if, for all $s$, $\boldsymbol{g}_s$ satisfies Equation 5, then $\boldsymbol{\Delta}_t$ satisfies it. Hereafter, we ignore the specific form of $\boldsymbol{\Delta}_t$, proving the result for a broader class of algorithms for which $\boldsymbol{\Delta}_t$ satisfies the OGD constraint. Continuing from Equation 10:

$$\frac{1}{n_o} \left( \sum_{i=1}^{n_o} \nabla_{\boldsymbol{\theta}} \boldsymbol{\Delta}_t^\intercal \boldsymbol{f_\theta}(\boldsymbol{x}_i) \left[ \nabla_{\boldsymbol{f}}^2 \ell_i \right] \nabla_{\boldsymbol{\theta}} \boldsymbol{f_\theta} \left( \boldsymbol{x}_i \right)^\intercal \boldsymbol{\Delta}_t \right)$$

$$\frac{1}{n_o} \left( \sum_{i=1}^{n_o} \nabla_{\boldsymbol{\theta}} \left[ \boldsymbol{\Delta}_t^\intercal \left[ \nabla_{\boldsymbol{\theta}} \boldsymbol{f}_{\boldsymbol{\theta}}^1(\boldsymbol{x}_i), \cdots, \nabla_{\boldsymbol{\theta}} \boldsymbol{f}_{\boldsymbol{\theta}}^K(\boldsymbol{x}_i) \right] \right] \left[ \nabla_{\boldsymbol{f}}^2 \ell_i \right] \nabla_{\boldsymbol{\theta}} \boldsymbol{f_\theta} \left( \boldsymbol{x}_i \right)^\intercal \boldsymbol{\Delta}_t \right)$$

$$\frac{1}{n_o} \left( \sum_{i=1}^{n_o} \nabla_{\boldsymbol{\theta}} \left[ \underbrace{\boldsymbol{\Delta}_t^\intercal \nabla_{\boldsymbol{\theta}} \boldsymbol{f}_{\boldsymbol{\theta}}^1(\boldsymbol{x}_i)}_{=0}, \cdots, \underbrace{\boldsymbol{\Delta}_t^\intercal \nabla_{\boldsymbol{\theta}} \boldsymbol{f}_{\boldsymbol{\theta}}^K(\boldsymbol{x}_i)}_{=0} \right] \left[ \nabla_{\boldsymbol{f}}^2 \ell_i \right] \nabla_{\boldsymbol{\theta}} \boldsymbol{f_\theta} \left( \boldsymbol{x}_i \right)^\intercal \boldsymbol{\Delta}_t \right) = 0,$$

where $\boldsymbol{\Delta}_t^\intercal \nabla_{\boldsymbol{\theta}} \boldsymbol{f}_{\boldsymbol{\theta}}^k(\boldsymbol{x}_i) = 0$ is the OGD constraint, which holds for any $k, i$ and $o < t$. This concludes the proof.

### A.1.5 COROLLARY 1

Consider an algorithm $\mathcal{A}$ satisfying Assumptions 1-2. By Theorem 3 we know that, if $\mathcal{A}(t) = \boldsymbol{\Delta}_t$ satisfies the OGD constraint, then $\boldsymbol{\Delta}_t^\intercal \boldsymbol{H}_o^\star \boldsymbol{\Delta}_t = 0 \, \forall o < t$. Consequently, $\boldsymbol{\Delta}_t^\intercal (\sum_{o=1}^{t-1} \boldsymbol{H}_o^\star) \boldsymbol{\Delta}_t = 0$. By Theorem 2, it follows that $\mathcal{E}(t) = 0 \, \forall t \in [T]$.

A.1.6 ON THE VALIDITY OF OGD-GTL

Instead of considering all the function gradients with respect to all the outputs $\{\nabla_{\boldsymbol{\theta}} \boldsymbol{f}_{\boldsymbol{\theta}}^k(\boldsymbol{x}) \,|\, k \in [1, K], \boldsymbol{x} \in D_o\}$, Farajtabar et al. (2020) also consider a cheaper approximation where they impose orthogonality only with respect to the index corresponding to the true ground truth label (GTL). We show this can be understood via fairly mild assumptions, if the loss function is cross-entropy.

For a cross-entropy loss, $\boldsymbol{f}_{\boldsymbol{\theta}}^k(\boldsymbol{x_i})$ is the log-probability (or logit) associated with class $k$ for input $\boldsymbol{x_i}$. The probability $p(y_i = k|\boldsymbol{x_i}; \boldsymbol{\theta})$ is then defined as $(\boldsymbol{p_i})_j = \mathrm{softmax}(\boldsymbol{f_{\theta}}(\boldsymbol{x_i}))_k$. Recall, from the proof of Theorem 3, that the theorem statement can be equivalently written as:

$$\boldsymbol{\Delta}_t^{\mathsf{T}} \left( \frac{1}{n_o} \sum_{i=1}^{n_o} \nabla_{\boldsymbol{\theta}} \boldsymbol{f_{\theta}}(\boldsymbol{x_i}) \left[ \nabla_{\boldsymbol{f}}^2 \ell_i \right] \nabla_{\boldsymbol{\theta}} \boldsymbol{f_{\theta}} \left( \boldsymbol{x_i} \right)^{\mathsf{T}} \right) \boldsymbol{\Delta}_t = 0$$

For a cross-entropy loss the Hessian of the loss with respect to the network output is given by:

$$\nabla_{\boldsymbol{f}}^2 \ell_i = \mathrm{diag}(\boldsymbol{p_i}) - \boldsymbol{p_i} \boldsymbol{p_i}^{\mathsf{T}}$$

Without loss of generality, assume index $k = 1$ corresponds to the GTL. Towards the end of the usual training, the softmax output at index 1, i.e., $p_1 \approx 1$. Let us assume that the probabilities for the remaining output coordinates is equally split between them. More precisely, let

$$p_2, \cdots, p_K = \frac{1 - p_1}{K - 1}.$$

Hence, in terms of their scales, we can regard $p_1 = \mathcal{O}(1)$ while $p_2, \cdots, p_K = \mathcal{O}\left(K^{-1}\right)$. Furthermore, $(1 - p_i) = \mathcal{O}(1) \quad \forall\, i \in [2 \cdots K]$. Then, a simple computation of the inner matrix in the Hessian outer product, i.e., $\nabla_{\boldsymbol{f}}^2 \ell = \mathrm{diag}(\boldsymbol{p}) - \boldsymbol{p}\boldsymbol{p}^{\top}$, will reveal that *diagonal entry corresponding to the ground truth label is $\mathcal{O}(1)$, while rest of the entries in the matrix are of the order $\mathcal{O}\left(K^{-1}\right)$ or $\mathcal{O}\left(K^{-2}\right)$* — thereby explaining this approximation. Also, we can see that this approximation would work well only when $K \gg 1$ and does not carry over to other loss functions, e.g. Mean Squared Error (MSE). In fact, $K = 2$ all the entries of this matrix are of the same magnitude $|p_1(1 - p_1)|$, and for MSE simply $\nabla_{\boldsymbol{f}}^2 \ell_i = \boldsymbol{I}_K$.

A.1.7 THEOREM 5

We start by proving Theorem 5. Before proceeding, we introduce the special notation to be used in the proof.

Notation: given some $a \times b$ matrix $\boldsymbol{C}$ we identify with $\mathrm{vec}(\boldsymbol{C})$ its $a \cdot b$ vector form ($\mathrm{vec}(\cdot)$ is the *vectorisation* operator). We use the standard *input×output* in order to determine the matrices dimensionality. Moreover, for simplicity, we consider all the network layers to have width $D$ (the same analysis can be done for layers of different widths). As an example, the gradient $\nabla_{\boldsymbol{W}^l} \mathcal{L}_t(\boldsymbol{\theta})$ is a tensor of dimensions $D \times D \times 1$, and $\mathrm{vec}(\nabla_{\boldsymbol{W}^l} \mathcal{L}_t(\boldsymbol{\theta}))$ is a vector of size $D^2$. Finally, we use $\boldsymbol{\theta}$ to refer to the *vector* of network parameters (the vectorization operation is implicit), thus $\nabla_{\boldsymbol{\theta}} \mathcal{L}_t(\boldsymbol{\theta})$, shortened to $\nabla \mathcal{L}_t(\boldsymbol{\theta})$ is a vector of size $P$.

Proof We recall the GPM algorithm. Let $\boldsymbol{x}_i^{l+1} = \sigma(\boldsymbol{W}^{l^{\mathsf{T}}} \boldsymbol{x}^l)$ be the representation of $\boldsymbol{x}_i$, by layer $l$ of the network ($\boldsymbol{x}_i^0 = \boldsymbol{x}_i$). Notice that the GPM algorithm does not allow for bias vectors in the layer functions, thus the set of network parameters is $\{\boldsymbol{W}^0, \ldots, \boldsymbol{W}^{L-1}\}$. The GPM constraint on the weight matrix update $\boldsymbol{\Delta}\boldsymbol{W}^l$ reads:

$$\langle \boldsymbol{\Delta}\boldsymbol{W}^l, \boldsymbol{x}_i^l \rangle = 0 \tag{11}$$

for all network layers $l$, $\boldsymbol{x}_i \in D_o$ and all previous tasks $\mathcal{T}_o$, $o < t$.

The optimization path from $\boldsymbol{\theta}_{t-1}$ to $\boldsymbol{\theta}_t$ is the sequence of parameter values $\{\boldsymbol{\theta}_{t-1 \to t}^1, \ldots, \boldsymbol{\theta}_{t-1 \to t}^{S_t}\}$, with parameter updates $\boldsymbol{\delta}_t^{(i)} = \boldsymbol{\theta}_{t-1 \to t}^i - \boldsymbol{\theta}_{t-1 \to t}^{i-1}$. In the following discussion we drop the task index and simply write $\boldsymbol{\delta}^{(i)} = \boldsymbol{\theta}^{(i)} - \boldsymbol{\theta}^{(i-1)}$. Similarly, $\boldsymbol{W}^{l^{(i)}} = \boldsymbol{W}^{l(i-1)} + \boldsymbol{\delta}_t^{(i)}[l]$ for the layer parameters. Observe that:

$$\mathrm{vec}(\nabla_{\boldsymbol{W}^l} \mathcal{L}_o(\boldsymbol{\theta}^{(i-1)}))^{\mathsf{T}} \, \mathrm{vec}(\boldsymbol{\delta}^{(i)}[l]) = 0 \,\, \forall\, l \implies \nabla \mathcal{L}_o(\boldsymbol{\theta}^{(i-1)})^{\mathsf{T}} \boldsymbol{\delta}^{(i)} = 0$$

and, similarly:

$$\text{vec}(\boldsymbol{\delta}_t^{(i)}[l])^\intercal \mathbf{H}_o(\boldsymbol{W}^{l(i-1)}) \,\text{vec}(\boldsymbol{\delta}_t^{(i)}[l]) = 0 \;\forall\, l$$

$$\implies \boldsymbol{\delta}_t^{(i)\,\intercal} \,\text{block-diag}(\mathbf{H}_o(\boldsymbol{\theta}_{t-1\to t}^{(i-1)}))\, \boldsymbol{\delta}_t^{(i)} = 0$$

Thus it is sufficient to prove the following two statements, for all $i \in [1, S_t]$, all layers $l$, and all $o < t$:

$$\text{vec}(\nabla_{\boldsymbol{W}^l}\mathcal{L}_o(\boldsymbol{\theta}^{(i-1)}))^\intercal \,\text{vec}(\boldsymbol{\delta}^{(i)}[l]) = 0 \tag{12}$$

$$\text{vec}(\boldsymbol{\delta}_t^{(i)}[l])^\intercal \mathbf{H}_o(\boldsymbol{W}^{l(i-1)}) \,\text{vec}(\boldsymbol{\delta}_t^{(i)}[l]) = 0 \tag{13}$$

By the linearity of the derivative $\nabla_{\boldsymbol{W}^l}\mathcal{L}_o(\boldsymbol{\theta}^{(i-1)}) = \frac{1}{n_o}\sum_{j=1}^{n_o}\nabla_{\boldsymbol{W}^l}\ell_j$, where $\ell_j = l_o(\boldsymbol{x_j}, y_j; \boldsymbol{\theta}^{(i-1)})$. We unpack $\nabla_{\boldsymbol{W}^l}\ell_j$ using the chain rule:

$$\nabla_{\boldsymbol{W}^l}\ell_j = \nabla_{\boldsymbol{W}^l}\boldsymbol{f}_{\boldsymbol{\theta}^{(i-1)}}(\boldsymbol{x_j}) \circ \nabla_{\boldsymbol{f}}\ell_j$$

Again, by the use of the chain rule we have:

$$\nabla_{\text{vec}(\boldsymbol{W}^l)}\boldsymbol{f}_{\boldsymbol{\theta}^{(i-1)}}(\boldsymbol{x_j}) = \nabla_{\text{vec}(\boldsymbol{W}^l)}\left(\boldsymbol{W}^{l(i-1)\,\intercal}\boldsymbol{x}_j^{l(i-1)}\right)\underbrace{\text{diag}\left(\sigma'(\boldsymbol{W}^{l(i-1)\,\intercal}\boldsymbol{x}_j^{l(i-1)})\right)}_{=:\boldsymbol{A}} \cdot \underbrace{\nabla_{\boldsymbol{x}_j^{l+1}}\boldsymbol{f}_{\boldsymbol{\theta}^{(i-1)}}(\boldsymbol{x_j})}_{=:\boldsymbol{\zeta}_{l+1}}$$

$$= \nabla_{\text{vec}(\boldsymbol{W}^l)}\left(\boldsymbol{W}^{l(i-1)\,\intercal}\boldsymbol{x}_j^{l(i-1)}\right)\cdot \boldsymbol{A}\cdot \boldsymbol{\zeta}_{l+1}$$

$$= (\boldsymbol{x}_j^{l(i-1)} \otimes \boldsymbol{I}_D)\cdot \boldsymbol{A}\cdot \boldsymbol{\zeta}_{l+1}$$

$$= (\boldsymbol{x}_j^{l(i-1)} \otimes \boldsymbol{A})\cdot \boldsymbol{\zeta}_{l+1}$$

where $\boldsymbol{A}$ contains the element-wise derivatives of the activation function $\sigma$ and $\boldsymbol{I}_D$ is the $D \times D$ identity matrix. Expanding recursively the layer activation:

$$\boldsymbol{x}_j^{l(i-1)} = \sigma(\boldsymbol{W}^{l-1(i-1)\,\intercal}\boldsymbol{x}_j^{l-1(i-1)})$$

$$\dots$$

$$\boldsymbol{x}_j^{1(i-1)} = \sigma(\boldsymbol{W}^{0(i-1)\,\intercal}\boldsymbol{x}_j)$$

As a consequence of the GPM constraint:

$$\boldsymbol{W}^{0(i-1)\,\intercal}\boldsymbol{x}_j = \left(\boldsymbol{W}_o^0 + (\boldsymbol{W}^{0(i-1)} - \boldsymbol{W}_o^0)\right)^\intercal \boldsymbol{x}_j = \boldsymbol{W}_o^{0\,\intercal}\boldsymbol{x}_j,$$

$\boldsymbol{W}_o^0$ being the value of $\boldsymbol{W}^0$ at the optimum of task $\mathcal{T}_o$, i.e. $\boldsymbol{\theta}_o$. By a recursive argument we then get:

$$\boldsymbol{x}_j^{1(i-1)} = \sigma(\boldsymbol{W}_o^{0\,\intercal}\boldsymbol{x}_j) = \boldsymbol{x}_j^1$$

$$\dots$$

$$\boldsymbol{x}_j^{l(i-1)} = \sigma(\boldsymbol{W}_o^{l-1\,\intercal}\boldsymbol{x}_j^{l-1}) = \boldsymbol{x}_j^l$$

In a nutshell, the GPM constraint freezes the network activation vectors to their value at the optimum, for each input stored in memory. Using this result, the expression of the output gradient simplifies to:

$$\nabla_{\text{vec}(\boldsymbol{W}^l)}\boldsymbol{f}_{\boldsymbol{\theta}^{(i-1)}}(\boldsymbol{x_j}) = (\boldsymbol{x}_j^l \otimes \boldsymbol{A})\cdot \boldsymbol{\zeta}_{l+1} \tag{14}$$

We are now ready finalize the first point of our proof (Equation 12).

$$\text{vec}(\nabla_{\boldsymbol{W}^l}\mathcal{L}_o(\boldsymbol{\theta}^{(i-1)}))^\intercal \,\text{vec}(\boldsymbol{\delta}^{(i)}[l]) =$$

$$(\nabla_{\text{vec}(\boldsymbol{W}^l)}\mathcal{L}_o(\boldsymbol{\theta}^{(i-1)}))^\intercal \,\text{vec}(\boldsymbol{\delta}^{(i)}[l]) =$$

$$\left(\frac{1}{n_o}\sum_{j=1}^{n_o}(\boldsymbol{x}_j^l \otimes \boldsymbol{A})\cdot \boldsymbol{\zeta}_{l+1}\cdot \nabla_{\boldsymbol{f}}\ell_j\right)^\intercal \,\text{vec}(\boldsymbol{\delta}^{(i)}[l]) =$$

$$\frac{1}{n_o}\sum_{j=1}^{n_o}\text{vec}(\boldsymbol{\delta}^{(i)}[l])^\intercal (\boldsymbol{x}_j^l \otimes \boldsymbol{A})\cdot \boldsymbol{\zeta}_{l+1}\cdot \nabla_{\boldsymbol{f}}\ell_j =$$

$$\frac{1}{n_o}\sum_{j=1}^{n_o}\text{vec}(\underbrace{\boldsymbol{x}_j^{l\,\intercal}\boldsymbol{\delta}^{(i)}[l]}_{=\boldsymbol{0}}\,\boldsymbol{A})\cdot \boldsymbol{\zeta}_{l+1}\cdot \nabla_{\boldsymbol{f}}\ell_j = 0 \qquad\qquad \text{(GPM constraint, eq. 11)}$$

We now proceed to the second point. As before, we decompose the Hessian into the sum of outer-product and functional Hessian:

$$\boldsymbol{H}_o(\boldsymbol{\theta}) = \frac{1}{n_o} \sum_{j=1}^{n_o} \nabla_{\boldsymbol{\theta}} \boldsymbol{f}_{\boldsymbol{\theta}}(\boldsymbol{x}_j) \left[ \nabla_{\boldsymbol{f}}^2 \ell_j \right] \nabla_{\boldsymbol{\theta}} \boldsymbol{f}_{\boldsymbol{\theta}}(\boldsymbol{x}_j)^{\mathsf{T}} + \frac{1}{n_o} \sum_{i=j}^{n_o} \sum_{k=1}^{K} [\nabla_{\boldsymbol{f}} \ell_j]_k \nabla_{\boldsymbol{\theta}}^2 \boldsymbol{f}_{\boldsymbol{\theta}}^k (\boldsymbol{x}_j)$$

In particular we are interested in the block-diagonal elements of the Hessian matrix, i.e. $\mathbf{H}_o(\boldsymbol{W}^{l(i-1)})$. When *ReLU* or linear activation functions are used the functional part does not contribute to the the the block-diagonal elements, as it can be seen from Equation 14:

$$\nabla_{\boldsymbol{W}^l}^2 \boldsymbol{f}_{\boldsymbol{\theta}^{(i-1)}}^k (\boldsymbol{x}_j) = \nabla_{\boldsymbol{W}^l} \left( (\boldsymbol{x}_j^l \otimes \boldsymbol{A}) \cdot \boldsymbol{\zeta}_{l+1} \right)$$
$$= (\boldsymbol{x}_j^l \otimes [\nabla_{\boldsymbol{W}^l} \boldsymbol{A}]) \cdot \boldsymbol{\zeta}_{l+1}$$

$\boldsymbol{A}$ is a diagonal matrix, and if $\sigma = \text{ReLU}$ then:

$$\boldsymbol{A}_{kk} = \begin{cases} 1 & \text{if } [\boldsymbol{W}^{l(i-1)^{\mathsf{T}}} \boldsymbol{x}_j^{l(i-1)}]_k > 0 \\ 0 & \text{otherwise} \end{cases}$$

Consequently, the gradient with respect to the layer weights $\boldsymbol{W}^{l(i-1)}$ is null. Similarly, if the activation function is linear, $\boldsymbol{A}$ is simply the identity matrix, and the gradient is 0. For a general activation function $\sigma$, the diagonal entries of $\boldsymbol{A}$ can have a non-linear dependence on $\boldsymbol{W}^{l(i-1)}$, and a case-by-case evaluation is required. Hereafter, we consider the piecewise-linear and linear case, for which the functional Hessian is *block-hollow*.

We can thus ignore the outer-product component in the Hessian layer blocks $\mathbf{H}_o(\boldsymbol{W}^{l(i-1)})$ and write:

$$\mathbf{H}_o(\boldsymbol{W}^{l(i-1)}) = \frac{1}{n_o} \sum_{j=1}^{n_o} \nabla_{\boldsymbol{W}^l} \boldsymbol{f}_{\boldsymbol{\theta}^{(i-1)}}(\boldsymbol{x}_j) \left[ \nabla_{\boldsymbol{f}}^2 \ell_j \right] \nabla_{\boldsymbol{W}^l} \boldsymbol{f}_{\boldsymbol{\theta}^{(i-1)}}(\boldsymbol{x}_j)^{\mathsf{T}}$$
$$= \frac{1}{n_o} \sum_{j=1}^{n_o} \left[ (\boldsymbol{x}_j^l \otimes \boldsymbol{A}) \cdot \boldsymbol{\zeta}_{l+1} \right] \left[ \nabla_{\boldsymbol{f}}^2 \ell_j \right] \left[ (\boldsymbol{x}_j^l \otimes \boldsymbol{A}) \cdot \boldsymbol{\zeta}_{l+1} \right]^{\mathsf{T}}$$

We can now finalise the second point in our proof (Equation 13).

$$\text{vec}(\boldsymbol{\delta}_t^{(i)}[l])^{\mathsf{T}} \mathbf{H}_o(\text{vec}(\boldsymbol{W}^{l(i-1)})) \, \text{vec}(\boldsymbol{\delta}_t^{(i)}[l]) =$$

$$= \text{vec}(\boldsymbol{\delta}_t^{(i)}[l])^{\mathsf{T}} \left( \frac{1}{n_o} \sum_{j=1}^{n_o} \left[ (\boldsymbol{x}_j^l \otimes \boldsymbol{A}) \cdot \boldsymbol{\zeta}_{l+1} \right] \left[ \nabla_{\boldsymbol{f}}^2 \ell_j \right] \left[ (\boldsymbol{x}_j^l \otimes \boldsymbol{A}) \cdot \boldsymbol{\zeta}_{l+1} \right]^{\mathsf{T}} \right) \text{vec}(\boldsymbol{\delta}_t^{(i)}[l]) =$$

$$= \frac{1}{n_o} \sum_{j=1}^{n_o} \left( \left[ \text{vec}(\boldsymbol{\delta}_t^{(i)}[l])^{\mathsf{T}} (\boldsymbol{x}_j^l \otimes \boldsymbol{A}) \cdot \boldsymbol{\zeta}_{l+1} \right] \right) \left[ \nabla_{\boldsymbol{f}}^2 \ell_j \right] \left( \left[ (\boldsymbol{x}_j^l \otimes \boldsymbol{A}) \cdot \boldsymbol{\zeta}_{l+1} \right]^{\mathsf{T}} \text{vec}(\boldsymbol{\delta}_t^{(i)}[l]) \right) =$$

$$= \frac{1}{n_o} \sum_{j=1}^{n_o} \left( \left[ \text{vec}(\underbrace{\boldsymbol{x}_j^{l^{\mathsf{T}}} \boldsymbol{\delta}_t^{(i)}[l]}_{=\mathbf{0}} \boldsymbol{A}) \cdot \boldsymbol{\zeta}_{l+1} \right] \right) \left[ \nabla_{\boldsymbol{f}}^2 \ell_j \right] \left( \left[ (\boldsymbol{x}_j^l \otimes \boldsymbol{A}) \cdot \boldsymbol{\zeta}_{l+1} \right]^{\mathsf{T}} \text{vec}(\boldsymbol{\delta}_t^{(i)}[l]) \right) = 0$$

Once more, $\boldsymbol{x}_j^{l^{\mathsf{T}}} \boldsymbol{\delta}_t^{(i)}[l]$ evaluates to 0 following the GPM constraint. We have evaluated only the right side of the expression (as it is enough to prove the equivalence), however notice that the expression is symmetric.

Notice that our proof applies to any $o < t$ and any $i \in [1, S_t]$ and layer $l$ in the network, since the GPM constraint also holds for all these cases. We have proved Theorem 5 for all network with piecewise-linear and linear activation functions. We leave the case of other activation functions to the reader.

### A.1.8   THEOREM 4

We are now ready to prove that, under Assumptions 1-2, algorithms satisfying the GPM constraint also satisfy the null-forgetting constraint (Equation 4, when the Hessian matrices are block diagonal.

Recall the null-forgetting constraint (Equation 4):

$$\mathcal{A}(t) = \boldsymbol{\Delta}_t = \arg\min \mathcal{E}(t) \iff \boldsymbol{\Delta}_t^\mathsf{T}(\sum_{o=1}^{t-1} \mathbf{H}_o^\star)\boldsymbol{\Delta}_t = 0$$

By Assumption 1, we know $\mathbf{H}_o^\star$ coincides with the outer-product Hessian:

$$\boldsymbol{H}_o(\boldsymbol{\theta}) = \frac{1}{n_o} \sum_{j=1}^{n_o} \nabla_{\boldsymbol{\theta}} \boldsymbol{f}_{\boldsymbol{\theta}_o}(\boldsymbol{x}_j) \left[\nabla_{\boldsymbol{f}}^2 \ell_j\right] \nabla_{\boldsymbol{\theta}} \boldsymbol{f}_{\boldsymbol{\theta}_o}(\boldsymbol{x}_j)^\mathsf{T}$$

Thus the statement to prove is:

$$\boldsymbol{\Delta}_t^\mathsf{T}(\sum_{o=1}^{t-1} \mathbf{H}_o^\star)\boldsymbol{\Delta}_t = 0$$

$$\iff \boldsymbol{\Delta}_t^\mathsf{T}(\sum_{o=1}^{t-1} \frac{1}{n_o} \sum_{j=1}^{n_o} \nabla_{\boldsymbol{\theta}} \boldsymbol{f}_{\boldsymbol{\theta}_o}(\boldsymbol{x}_j) \left[\nabla_{\boldsymbol{f}}^2 \ell_j\right] \nabla_{\boldsymbol{\theta}} \boldsymbol{f}_{\boldsymbol{\theta}_o}(\boldsymbol{x}_j)^\mathsf{T})\boldsymbol{\Delta}_t = 0$$

$$\iff \sum_{o=1}^{t-1} \frac{1}{n_o} \sum_{j=1}^{n_o} \boldsymbol{\Delta}_t^\mathsf{T} \left(\nabla_{\boldsymbol{\theta}} \boldsymbol{f}_{\boldsymbol{\theta}_o}(\boldsymbol{x}_j) \left[\nabla_{\boldsymbol{f}}^2 \ell_j\right] \nabla_{\boldsymbol{\theta}} \boldsymbol{f}_{\boldsymbol{\theta}_o}(\boldsymbol{x}_j)^\mathsf{T}\right) \boldsymbol{\Delta}_t = 0$$

$$\iff \sum_{o=1}^{t-1} \frac{1}{n_o} \sum_{j=1}^{n_o} (\sum_{i=1}^{S_t} \boldsymbol{\delta}_t^{(i)})^\mathsf{T} \left(\nabla_{\boldsymbol{\theta}} \boldsymbol{f}_{\boldsymbol{\theta}_o}(\boldsymbol{x}_j) \left[\nabla_{\boldsymbol{f}}^2 \ell_j\right] \nabla_{\boldsymbol{\theta}} \boldsymbol{f}_{\boldsymbol{\theta}_o}(\boldsymbol{x}_j)^\mathsf{T}\right) (\sum_{i=1}^{S_t} \boldsymbol{\delta}_t^{(i)}) = 0$$

$$\iff \sum_{o=1}^{t-1} \frac{1}{n_o} \sum_{j=1}^{n_o} \left(\sum_{i=1}^{S_t} \boldsymbol{\delta}_t^{(i)\mathsf{T}} \nabla_{\boldsymbol{\theta}} \boldsymbol{f}_{\boldsymbol{\theta}_o}(\boldsymbol{x}_j)\right) \left[\nabla_{\boldsymbol{f}}^2 \ell_j\right] \left(\sum_{i=1}^{S_t} \nabla_{\boldsymbol{\theta}} \boldsymbol{f}_{\boldsymbol{\theta}_o}(\boldsymbol{x}_j)^\mathsf{T} \boldsymbol{\delta}_t^{(i)}\right) = 0$$

Being the Hessian matrix block-diagonal by assumption, it is sufficient to prove:

$$\text{vec}(\boldsymbol{\delta}_t^{(i)}[l])^\mathsf{T} \nabla_{\boldsymbol{W}^l} \boldsymbol{f}_{\boldsymbol{\theta}_o}(\boldsymbol{x}_j) = 0$$

where we adopt the notation from the proof of Theorem 5. Using the result in Equation 14:

$$\text{vec}(\boldsymbol{\delta}_t^{(i)}[l])^\mathsf{T} \nabla_{\boldsymbol{W}^l} \boldsymbol{f}_{\boldsymbol{\theta}_o}(\boldsymbol{x}_j) = \text{vec}(\boldsymbol{\delta}_t^{(i)}[l])^\mathsf{T} (\boldsymbol{x}_j^l \otimes \boldsymbol{A}) \cdot \boldsymbol{\zeta}_{l+1}$$

$$= \text{vec}(\boldsymbol{x}_j^{l\mathsf{T}} \boldsymbol{\delta}_t^{(i)}[l] \boldsymbol{A}) \cdot \boldsymbol{\zeta}_{l+1} = 0,$$

where the last equality follows from the GPM constraint.

### A.1.9 COROLLARY 2

Finally we go over the proof of Corollary 2. We start by considering the following Taylor expansion:

$$\mathcal{E}_o(\boldsymbol{\theta}_{t-1 \to t}^{(i)}) = \mathcal{E}_o(\boldsymbol{\theta}_{t-1 \to t}^{(i-1)}) + \nabla \mathcal{L}_o(\boldsymbol{\theta}_{t-1 \to t}^{(i-1)})^\mathsf{T} \boldsymbol{\delta}_t^{(i)} + \frac{1}{2} \boldsymbol{\delta}_t^{(i)\mathsf{T}} \mathbf{H}_o(\boldsymbol{\theta}_{t-1 \to t}^{(i-1)}) \boldsymbol{\delta}_t^{(i)} + \mathcal{O}(\|\boldsymbol{\delta}_t^{(i)}\|)$$

By the GPM constraint, $\nabla \mathcal{L}_o(\boldsymbol{\theta}_{t-1 \to t}^{(i-1)})^\mathsf{T} \boldsymbol{\delta}_t^{(i)} = 0$ and $\boldsymbol{\delta}_t^{(i)\mathsf{T}} \mathbf{H}_o(\boldsymbol{\theta}_{t-1 \to t}^{(i-1)}) \boldsymbol{\delta}_t^{(i)} = 0$, as by assumption block-diag$(\mathbf{H}_o(\boldsymbol{\theta}_{t-1 \to t}^{(i-1)}) = \mathbf{H}_o(\boldsymbol{\theta}_{t-1 \to t}^{(i-1)})$. Consequently, the expression of forgetting on the optimization path simplifies to:

$$\mathcal{E}_o(\boldsymbol{\theta}_{t-1 \to t}^{(i)}) = \mathcal{E}_o(\boldsymbol{\theta}_{t-1 \to t}^{(i-1)}) + \mathcal{O}(\|\boldsymbol{\delta}_t^{(i)}\|)$$

For sufficiently small step sizes (often $\|\boldsymbol{\delta}_t^{(i)}\| \in \mathcal{O}(\eta)$, where $\eta$ is the learning rate), the approximation error is negligible, and, by recursively applying the above identity we obtain: $\mathcal{E}_o(\boldsymbol{\theta}_t) = \mathcal{E}_o(\boldsymbol{\theta}_o) = 0$, which proves the statement.

### A.2 FURTHER DISCUSSION

### A.2.1 NON MONOTONIC NULL-FORGETTING

The null-forgetting constraint presupposes that forgetting is zero after learning each task, i.e. $\mathcal{E}_\tau(t) = 0$ for any $\tau$ and $t$ (with $\tau < t$). Here we consider what happens when $\mathcal{E}(\tau) > 0$ for some $\tau < t$. Specifically, we want to derive a new constraint on the update $\boldsymbol{\Delta}_{\tau+1}$ such that $\mathcal{E}(\tau + 1) = 0$.

**Theorem 6** (Non monotonic null-forgetting constraint.)**.** *Let $\mathcal{A} : [T] \to \Theta$ be a continual learning algorithm satisfying Assumptions 1-2. Furthermore, for the first $\tau - 1$ tasks it holds that $\mathcal{E}(k) = 0$. Let $\mathcal{A}(\tau) = \mathbf{\Delta}_\tau$ s.t. $\mathcal{E}(\tau) > 0$.*

*Then for an update $\mathcal{A}(\tau + 1) = \mathbf{\Delta}_{\tau+1}$, $\mathcal{E}(\tau + 1) = 0$ if and only if:*

$$\mathbf{\Delta}_{\tau+1} = \boldsymbol{z} - (\sum_{o=1}^{\tau} \mathbf{H}_o^\star)^\dagger \boldsymbol{B} \mathbf{\Delta}_\tau, \tag{16}$$

*for any vector $\boldsymbol{z}$ satisfying the condition $(\sum_{o=1}^{\tau} \mathbf{H}_o^\star)\boldsymbol{z} = \mathbf{0}$. $\boldsymbol{B}$ is the matrix characterising $\mathcal{E}(\tau)$, i.e. $\mathcal{E}(\tau) = \frac{1}{2\tau}\mathbf{\Delta}_\tau^\intercal \boldsymbol{B} \mathbf{\Delta}_\tau$.*

Proof. By Assumptions 1-2 we can approximate $\mathcal{E}(\tau + 1)$ using Equation 3:

$$\mathcal{E}(\tau+1) = \frac{1}{\tau+1}\left(\tau \cdot \mathcal{E}(\tau) + \frac{1}{2}\mathbf{\Delta}_{\tau+1}^\intercal(\sum_{o=1}^{\tau}\mathbf{H}_o^\star)\mathbf{\Delta}_{\tau+1} + \sum_{o=1}^{\tau-1}(\boldsymbol{\theta}_\tau - \boldsymbol{\theta}_o)^\intercal\mathbf{H}_o^\star\mathbf{\Delta}_{\tau+1}\right)$$

Using $\mathcal{E}(k) = 0 \,\forall\, k < \tau$ and Theorem 1 we have $\mathcal{E}(\tau) = \mathbf{\Delta}_\tau^\intercal \boldsymbol{B}\mathbf{\Delta}_\tau = C > 0$, where we define $\boldsymbol{B} := (\sum_{o=1}^{\tau-1}\mathbf{H}_o^\star)$. Moreover, notice that:

$$\boldsymbol{v}_\tau := \sum_{o=1}^{\tau-1}(\boldsymbol{\theta}_\tau - \boldsymbol{\theta}_o)^\intercal\mathbf{H}_o = \sum_{o=1}^{\tau-1}\mathbf{\Delta}_\tau^\intercal\mathbf{H}_o^\star + \boldsymbol{v}_{\tau-1}$$

From Theorem 1 it follows that $\boldsymbol{v}_{\tau-1} = \mathbf{0}$ and thus:

$$\mathcal{E}(\tau+1) = \frac{1}{\tau+1}\left(\tau \cdot C + \frac{1}{2}\mathbf{\Delta}_{\tau+1}^\intercal(\sum_{o=1}^{\tau}\mathbf{H}_o^\star)\mathbf{\Delta}_{\tau+1} + \mathbf{\Delta}_\tau^\intercal\boldsymbol{B}\mathbf{\Delta}_{\tau+1}\right)$$

Notice that this expression is quadratic in $\mathbf{\Delta}_{\tau+1}$ and thus $\mathcal{E}(\tau + 1)$ admits a closed-form solution. We get:

$$\mathrm{argmin}_{\mathbf{\Delta}_{\tau+1}}\mathcal{E}(\tau+1) = \boldsymbol{z} - (\sum_{o=1}^{\tau}\mathbf{H}_o^\star)^\dagger\boldsymbol{B}\mathbf{\Delta}_\tau,$$

where $\boldsymbol{z}$ is any vector satisfying the condition $(\sum_{o=1}^{\tau}\mathbf{H}_o^\star)\boldsymbol{z} = \mathbf{0}$. Notice that if $\mathcal{E}(\tau) = 0$ then $\boldsymbol{B}\mathbf{\Delta}_\tau = \mathbf{0}$ and 6 reduces to the null-forgetting constraint from Theorem 2. Intuitively, in order to reverse forgetting, the algorithm has to retrace its steps along the harmful directions in the previous update (i.e. $\boldsymbol{B}\mathbf{\Delta}_\tau$).

The next Corollary extends the result in Theorem 1 to the case of non-monotonic forgetting. It follows directly from Theorem 6 and the formulation of forgetting in Equation 3.

**Corollary 3** (Non monotonic null-forgetting.)**.** *For any continual learning algorithm satisfying Assumptions 1-2 the following relationship between previous and current forgetting exists:*

$$\mathcal{E}(1), \ldots, \mathcal{E}(t-2) = 0, \mathcal{E}(t-1) > 0, \mathcal{E}(t) = 0 \implies \mathcal{E}(t+2) = \frac{1}{2}\mathbf{\Delta}_{t+2}^\intercal\left(\frac{1}{t+2}\cdot\sum_{o=1}^{t+1}\mathbf{H}_o^\star\right)\mathbf{\Delta}_{t+2}$$

### A.2.2 IMPLICATIONS OF NULL-FORGETTING

Theorem 1 describes forgetting in terms of the alignment between the current parameter update and the column space of the Hessians of the previous tasks.

Let $\overline{\mathbf{H}}_{<t}^\star := \frac{1}{t-1}\cdot\sum_{o=1}^{t-1}\mathbf{H}_o^\star$ denote the average Hessian matrix of the old tasks. Intuitively, the higher the rank of $\overline{\mathbf{H}}_{<t}^\star$ the less capacity is left to learn a new task. In order to see this, simply consider the constrained optimization problem:

$$\mathbf{\Delta}_t = \mathrm{argmin}_{\mathbf{\Delta}\in\Theta}\,\mathcal{L}_t(\mathbf{\Delta} + \boldsymbol{\theta}_{t-1})$$
$$s.t.\,\mathbf{\Delta}_t^\intercal\overline{\mathbf{H}}_{<t}^\star\mathbf{\Delta}_t = 0$$

The constraint on the parameter update effectively restricts the search space from $\Theta$ to the $P - R$-dimensional subspace orthogonal to the column space of $\overline{\mathbf{H}}_{<t}^\star$, where $R = \mathrm{rank}(\overline{\mathbf{H}}_{<t}^\star)$. Thus, the

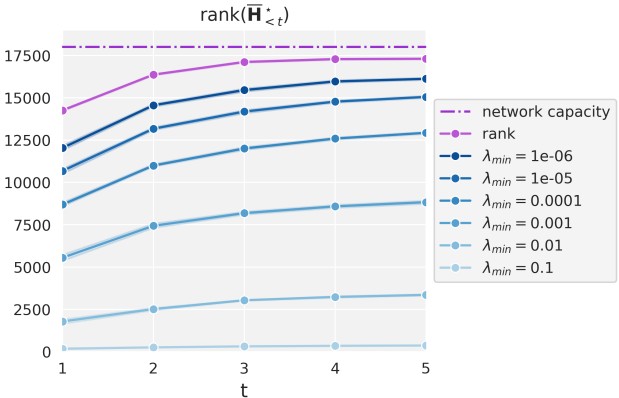

Figure 5: Rank and effective rank for various threshold $\lambda$ values on the Rotated MNIST challenge, learned by SGD. The values are averaged over seeds.

higher $R$, the smaller the set of feasible solutions for the current task. Importantly, this formulation ignores potential *constructive interference* between tasks, which is a limitation of many parameter isolation models (Lin et al. (2022) do follow-up work on GPM addressing this problem). This condition is necessary under Assumption 1-2, where forgetting is always non-negative and thus there can be no constructive interference. However, in the more general setting, the condition reflects a worse-case scenario, and exploiting a constructive intereference of task may save network capacity for future tasks.

Methods enforcing sparsity on the parameter updates (Schwarz et al., 2021) induce a transformation of the loss gradient vector and Hessian matrix similar to parameter isolation methods. In practice, the column space of $\boldsymbol{H}_o$ is a subset of the task parameters $\Theta_o$. Thereby, the alignment of the current update to the column space of the average Hessian becomes simply a function of the overlap with the previous updates $\|\boldsymbol{\Delta}_t - \boldsymbol{\theta}_{t-1}\|_0$ and the average Hessian rank is $R \leq (\sum_{o=1}^{t-1} r_o) \cdot P$, where $r_o$ is the fraction of active connections for task $\mathcal{T}_o$. If the average Hessian rank is not maximal, Theorem 1 suggests that $r_t > (1 - (\sum_{o=1}^{t-1} r_o))$ can be achieved without suffering forgetting.

In Figure 5 we plot the evolution of $\mathrm{rank}(\overline{\mathbf{H}}^{\star}_{<t})$ over $t$ for the Rotated-MNIST challenge. Additionally, we evaluate the effective rank of the average Hessian for multiple threshold $\lambda$ values, which better captures the effective dimensionality of the matrix column space.

# B EXPERIMENTAL DETAILS

## B.0.1 EXPERIMENTAL SETUP

The experiments are designed to support the theoretical contribution of the paper. The specific experimental design choices are motivated by the theory and by the common practices in the field. Especially when providing results on existing methods (Section 5.2), we employ similar experimental setting as in the original paper. Following is a detailed description of our experimental setup, which is aimed at facilitating reproducibility.

**Datasets.**

1. *Rotated-MNIST*: a sequence of 5 tasks, each corresponding to a fixed rotation applied to the MNIST dataset. We fix the set of rotations to $[-45°, -22.5°, 0, 22.5°, 45°]$, in line with (Farajtabar et al., 2020). In order to be able to compute the network Hessian matrix, the input is downscaled to $14 \times 14$. The output space is shared across task, and has 10 dimensions, one for each digit. We employ the standard Pytorch train-test split, loading $60,000$ inputs to train each task.

2. *Split CIFAR-10*: a sequence of 5 tasks, each corresponding to a different pair of labels in the CIFAR-10 dataset. The splits are fixed across experiments: classes $(0, 1)$ for the first task, $(2, 3)$ for the second one, and so on. The output space is shared across task, and has

      dimensionality 2. We employ the standard Pytorch train-test split, loading $5,000 \times 2$ inputs to train each task.

3. *Split CIFAR-100*: a sequence of 20 tasks. Similarly to Split CIFARO-10, each task corresponds to a different set of 5 classes from the CIFAR-100 dataset. The splits are fixed across experiments. The output space is shared across task, and has dimensionality 5. We employ the standard Pytorch train-test split, loading $500 \times 5$ inputs to train each task.

**Network architectures.** Overall, we employ 3 different network architectures in the experiments: an MLP, a CNN, and a Transformer. We here describe the implementation of the three networks. In Table 3 we provide an overview of the different hyperparameter settings used: each experiment configuration takes one row of the table.

1. The MLP has 3 hidden layers with 50 units each. Experiments with similar MLP architectures are widespread in the literature (Farajtabar et al., 2020; Saha et al., 2021; Mirzadeh et al., 2020b; Hsu et al., 2018). However, we reduce the conventional layer width from 100 (or 200) to 50 neurons, while increasing the depth by one layer (from 2 to 3), thereby lowering the network size. Our network has a total of $18,010$ parameters. In the GPM experiments we do not include bias terms in the network, thus reducing the size further to $17,800$. We train this network using SGD, SGD$^\dagger$, OGD and GPM with the same hyperparameter configurations.

2. The CNN used is a ResNet 18 (He et al., 2016). We adopt the same implementation as (Mirzadeh et al., 2020b; Saha et al., 2021; Lopez-Paz & Ranzato, 2017), which is a smaller version of the original ResNet18, with three times less feature maps per layer. In total, the network has around 0.5M parameters. This network is trained using SGD and SGD$^\dagger$ on Split CIFAR-10 and Split CIFAR-100 (the output layer size is changed accordingly). The same hyperparameter configuration is used for both SGD and SGD$^\dagger$ in order to make a fair comparison.

3. The Transformer is a ViT (Dosovitskiy et al., 2020). The use of Transformers is not as widespread yet in the continual learning literature. We refer to a publicly available implementation of a ViT in Pytorch[1], which is again a smaller version than the original. The network has around 12M parameters. We retain the standard configuration provided in the reference implementation for almost all hyperparameters, except we simplify the optimization scheme. We use SGD instead of the Adam optimizer, no weight decay or label smoothing and a simple two-steps learning rate decay schedule. We train this network for SGD and SGD$^\dagger$ again on Split CIFAR-10 and Split CIFAR-100, changing the output layer accordingly.

**Metrics.** Finally, we recapitulate the metrics used in the paper. We measure forgetting on a specific task as $\mathcal{E}_o(t) = \mathcal{L}_o(\boldsymbol{\theta}_t) - \mathcal{L}_o^\star$ and average forgetting as $\mathcal{E}(t) = \frac{1}{t}\sum_{o=1}^{t-1}\mathcal{E}_o(t)$. In the literature, forgetting is often measured as $\mathrm{BWT} = \frac{1}{T-1}\sum_{o=1}^{T-1}(a_{o,o} - a_{T,o})$, where $a_{j,i}$ denotes the accuracy on task $\mathcal{T}_i$ for $\boldsymbol{\theta} = \boldsymbol{\theta}_j$. In our results we also report forgetting in terms of accuracy, in order to increase the interpretability of the scores. Simply, we replace the definition of forgetting with $\mathcal{E}_o(t) = a_{o,o} - a_{t,o}$ and $\mathcal{E}(t) = \frac{1}{t}\sum_{o=1}^{t-1}\mathcal{E}_o^a(t)$. Finally, in accordance with the convention in the literature, we also measure the average accuracy over tasks as $a(t) = \frac{1}{t}\sum_{o=1}^{t}a_{t,o}$. For all these scores we evaluate the loss and accuracy on the test set.

**Selection of $k$ in experiment 5.2** When comparing OGD, GPM and SGD† we have to be careful in selecting the number of new vectors to store in the memory for each task, since each method uses different vectors in practice. We decide to threshold the spectral energy of the addition to the memory, using $\epsilon$ as a cutoff parameter. In practice, instead of controlling , the number of eigenvectors, we control , the portion of spectral energy left out, where spectral energy is the squared eigenvalues norm . From the eigendecomposition, we obtain the cumulative spectral energy and we make a cut keeping $1 - \epsilon$ of the total energy. In this way, the size of the 'protected subspace' depends on the task, and in particular on the geometry of the optimum.

---

[1]Found on Github: https://github.com/omihub777/ViT-CIFAR. Last accessed in May 2023.

Table 3: Experiments configurations

| Network | Dataset | Algorithm | Learning rate | Epochs |
|---|---|---|---|---|
| MLP | Rotated-MNIST | SGD | 0.01 | 5 |
| | | SGD | $\{0.01, 10^{-5}\}$ | 15 |
| | | SGD$^\dagger$, $\epsilon = 0.01$ | 0.01 | 5 |
| | | SGD$^\dagger$, $\epsilon = 0.01$ | $\{0.01, 10^{-5}\}$ | 15 |
| | | GPM, $\epsilon = 0.01$ | 0.01 | 5 |
| | | GPM, $\epsilon = 0.01$ | $\{0.01, 10^{-5}\}$ | 15 |
| | | OGD-gtl, $\epsilon = 0.01$ | 0.01 | 5 |
| | | OGD-gtl, $\epsilon = 0.01$ | $\{0.01, 10^{-5}\}$ | 15 |
| ResNet18 | Split CIFAR-10 | SGD | $\{0.1, 0.001\}$ | $\{10/20, 1\}$ |
| | | SGD$^\dagger$, $k = 10$ | $\{0.1, 0.001\}$ | $\{10/20, 1\}$ |
| | | SGD$^\dagger$, $k = 20$ | $\{0.1, 0.001\}$ | $\{10/20, 1\}$ |
| | Split CIFAR-100 | SGD | $\{0.1, 0.001\}$ | $\{30/60/70, 10\}$ |
| | | SGD$^\dagger$, $k = 10$ | $\{0.1, 0.001\}$ | $\{30/60/70, 10\}$ |
| | | SGD$^\dagger$, $k = 20$ | $\{0.1, 0.001\}$ | $\{30/60/70, 10\}$ |
| ViT | Split CIFAR-10 | SGD | $\{0.1, 0.001\}$ | $\{10/20, 1\}$ |
| | | SGD$^\dagger$, $k = 10$ | $\{0.1, 0.001\}$ | $\{10/20, 1\}$ |
| | | SGD$^\dagger$, $k = 20$ | $\{0.1, 0.001\}$ | $\{10/20, 1\}$ |
| | Split CIFAR-100 | SGD | $\{0.1, 0.001\}$ | $\{15/25, 5/10\}$ |
| | | SGD$^\dagger$, $k = 10$ | $\{0.1, 0.001\}$ | $\{15/25, 5/10\}$ |
| | | SGD$^\dagger$, $k = 20$ | $\{0.1, 0.001\}$ | $\{15/25, 5/10\}$ |

Table 4: All the experiments in the table are repeated over 5 seeds, namely $11, 13, 21, 33, 55$. When the same learning rate is used for all the tasks it is written as a single number. Otherwise, we use a different learning rate for the tasks following the first and we indicate this as $\{r_1, r_{2:T}\}$ in the table. The batch size is 10 unless stated otherwise. The epoch number follows a similar scheme: if a different number of epochs is used between the first and remaining tasks we write $\{e_1, e_{2:T}\}$. Moreover, if the learning rate is decayed while learning a task we write the decay schedule in the place of the epoch number as $\{s_1^1/s_1^2/\ldots/e_1, s_{2:T}^1/s_{2:T}^2/\ldots/e_{2:T}\}$, which means that the first learning learning rate is decayed after $s_1^1$ epochs and then again $s_1^2$, until reaching the final epoch $e_1$. Similarly, for the remaining tasks, the learning rate is decayed after $s_{2:T}^1$ epochs and then again $s_{2:T}^2$, and so on until reaching the final epoch $e_{2:T}$.

### B.0.2 HESSIAN COMPUTATION AND EIGENDECOMPOSITION

The loss Hessian matrix plays a key role in our theoretical framework, and thus its computation is necessary to our experiments. However, the Hessian matrix has a high memory cost, i.e. $\mathcal{O}(P^2)$ ($P$ being the size of the network). Moreover, SGD$^\dagger$ uses the principal eigenvectors of the Hessian, and the eigendecomposition operation has complexity $\mathcal{O}(P^3)$. For the MLP network we can compute the full Hessian matrix in $\approx 160s$ and its eigendecomposition in $\approx 960s$ For the CNN and the Transformer networks, we cannot compute the full Hessian matrix. However, we can still obtain its principal eigenvectors through iterative methods such as *power iteration* or the *Lanczos method* and the Hessian-vector product (Yao et al., 2018a; Xu et al., 2018). Golmant et al. (2018) has publicly provided an implementation of these methods for Pytorch neural networks, which we have adapted to our scope. We use a random subset of the data to compute the Hessian and/or its eigenvectors. For all the experiments we fix the subset size to 1000 samples.

# C  ADDITIONAL RESULTS

## C.1  EXTENDED RESULTS

In this section we present the results of the experiments in their entirety. We adopt the same structure of Section 5 of the paper.

### C.1.1  EXPERIMENTS SECTION 5.2

With the first set of experiments, we investigate the empirical validity of Theorems 3-4. In the paper, we show the equivalence of GPM, OGD and SGD$^{\dagger}$ when the analysis assumptions are met. Here we further examine GPM and OGD.

**Measuring the violation of the constraint**  Corollary 1 and 2 establish zero forgetting guarantees for OGD and GPM under Assumption 1-2. We directly measure the degree of violation of the null-forgetting constraint (Equation 4) on the RMNIST dataset for both methods and compare them with vanilla SGD. The score used is simply:

$$VNC(t) = \mathbf{\Delta}_t^{\intercal}(\frac{1}{t}\sum_{o=1}^{t-1}\mathbf{H}_o^{\star})\mathbf{\Delta}_t$$

The null-forgetting constraint is satisfied if $VNC(t) = 0$ for all $t$. The results, reported in Table 5, agree with our theoretical intuition: GPM and OGD consisently achieve lower degrees of violation of the null-forgetting constraint.

Table 5: Violation of null-forgetting constraint (Equation 4).

| lr | Algorithm | $VNC(t)$ |
|---|---|---|
| $1e^{-5}$ | GPM | $0.004 \pm 0.004$ |
| | OGD | $\mathbf{0.0001} \pm 0.0004$ |
| | SGD | $0.27 \pm 0.10$ |
| $1e^{-2}$ | GPM | $\mathbf{0.14} \pm 0.11$ |
| | OGD | $0.25 \pm 0.05$ |
| | SGD | $0.59 \pm 0.07$ |

**Verifying the underlying assumptions**  Next, we check the underlying assumptions of both GPM and OGD. Through our analysis we show that GPM relies on a block-diagonal approximation of the loss Hessian matrix. In order to evaluate the validity of the assumption we define a *block-diagonality* score:

$$\mathfrak{D}(M) = 1 - \frac{\mathbb{E}_{(i,j)\notin D}|M_{ij}|}{\mathbb{E}_{(i,j)\in D}|M_{ij}|},$$

where we use $B$ to denote the set of entries in the diagonal blocks, and $M$ is any matrix. Notice that a perfectly block-diagonal matrix will score $\mathfrak{D}(M) = 1$. For non block-diagonal matrices the score has no lower bound. On random matrices, the score evaluates to $-0.001\,(\pm 0.02)$, over 1000 trials. In Table 6 (left), we record the average score for the Hessian matrix $\boldsymbol{H}_t^{\star}$ of different tasks when training with GPM, both in the high and low learning rate cases. The average is taken over seeds and tasks. Importantly, the results reveal that the assumption is not met in practice.

OGD effectively adopts the outer-product Hessian $\boldsymbol{H}_t^{O}$ (Equation 9), which has been shown to be a good approximation of the Hessian matrix towards the end of training (Singh et al., 2021) - and thus is automatically satisfied under Assumption 1. We verify this assumption by computing the *spectral similarity* of the Hessian and its outer-product component. We define the spectral similarity between two matrices as follows:

$$\mathfrak{S}(\boldsymbol{A}, \boldsymbol{B}) = \frac{\langle \boldsymbol{A}, \boldsymbol{B} \rangle_F}{\|\boldsymbol{A}\|_F \|\boldsymbol{B}\|_F},$$

where $\langle \cdot, \cdot \rangle_F$ denotes the Frobenius inner product and $\| \cdot \|_F$ denotes the Frobenius norm. This score is equal to 1 for two equivalent matrices, while, for dissimilar matrices it can be arbitrarily low. We

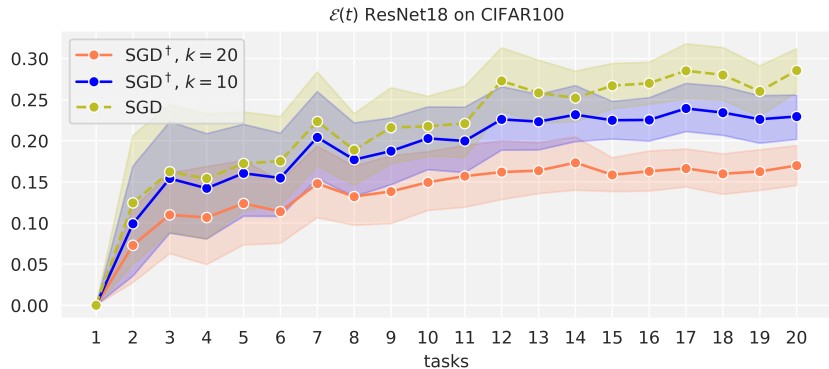

Figure 6: *Convolutional networks*: Average forgetting $\mathcal{E}(t)$ of SGD and SGD$^\dagger$ with varying values of $k$ on Split CIFAR-100 for ResNet-18.

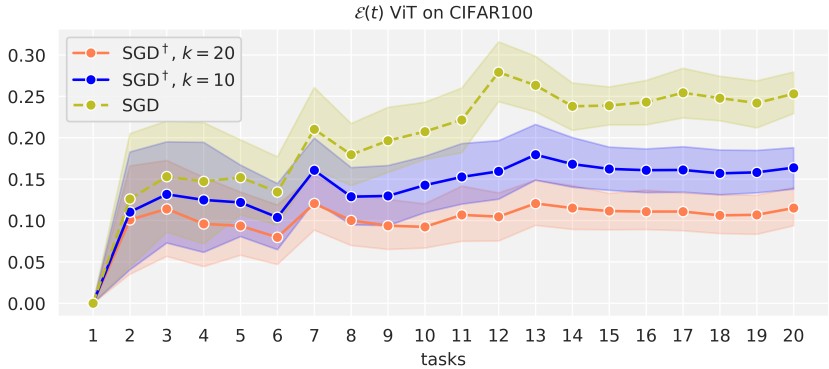

Figure 7: *Vision Transformers*: Average forgetting $\mathcal{E}(t)$ of SGD and SGD$^\dagger$ with varying values of $k$ on Split CIFAR-100 for Vision Transformer.

test the score on pairs of random matrices, and obtain an average of $-0.0002\,(\pm 0.01)$ over 1000 trials. In Table 6 (right) we report the score $\mathfrak{S}(\boldsymbol{H}_t^O, \boldsymbol{H}_t^\star)$, averaged over tasks and seeds. For both the high and low learning rate case the score is around $0.7$.

Table 6: Validity of the assumptions underlying GPM (left) and OGD (right).

| lr | $\mathfrak{D}(\boldsymbol{H}_t^\star)$ | lr | $\mathfrak{S}(\boldsymbol{H}_t^O, \boldsymbol{H}_t^\star)$ |
|---|---|---|---|
| $1e^{-5}$ | $-0.04 \pm 0.12$ | $1e^{-5}$ | $0.70 \pm 0.08$ |
| $1e^{-2}$ | $-0.11 \pm 0.13$ | $1e^{-2}$ | $0.69 \pm 0.02$ |

### C.1.2   EXPERIMENTS SECTION 5.3

The second set of experiments is aimed at testing the validity of our analysis on other architectures. We employ CNNs and Transformers, as described in Section B. Figure 6 and 7 show the average forgetting on Split CIFAR-100 for tasks 1 to 20 (previous results only considered the first 10 tasks for brevity). Moreover, we could compute up to the first 40 eigenvectors for the ResNet, and we include the results in Figure 6.

Next, we report a comprehensive view of the results on Split CIFAR-10 in Table 7. Similarly to Split CIFAR-100, we observe the average forgetting increasing overall with $t$. Importantly, both the CNN and the Transformer record significantly lower levels of forgetting when trained with SGD$^\dagger$, once again supporting the validity of our theoretical results.

Table 7: $\mathcal{E}(t)$ on Split CIFAR-10.

|  | Algorithm | $\mathcal{E}(2)$ | $\mathcal{E}(3)$ | $\mathcal{E}(4)$ | $\mathcal{E}(5)$ |
|---|---|---|---|---|---|
| ResNet | SGD | $0.0127 \pm 0.0200$ | $0.0186 \pm 0.0196$ | $0.0375 \pm 0.0515$ | $0.0285 \pm 0.0267$ |
|  | SGD$^\dagger$-10 | $0.0100 \pm 0.0177$ | $0.0092 \pm 0.0145$ | $0.0201 \pm 0.0198$ | $0.0261 \pm 0.0313$ |
|  | SGD$^\dagger$-20 | $\mathbf{0.0094} \pm 0.0169$ | $\mathbf{0.0076} \pm 0.0167$ | $\mathbf{0.0171} \pm 0.0172$ | $\mathbf{0.0213} \pm 0.0262$ |
| ViT | SGD | $0.0306 \pm 0.0329$ | $0.0249 \pm 0.0383$ | $0.0476 \pm 0.0669$ | $0.0264 \pm 0.0282$ |
|  | SGD$^\dagger$-10 | $\mathbf{0.0122} \pm 0.0153$ | $0.0073 \pm 0.0147$ | $0.0081 \pm 0.0161$ | $0.0057 \pm 0.0118$ |
|  | SGD$^\dagger$-20 | $0.0126 \pm 0.0173$ | $\mathbf{0.0061} \pm 0.0160$ | $\mathbf{0.0077} \pm 0.0154$ | $\mathbf{0.0048} \pm 0.0110$ |

Additionally, we report average accuracy on both Split CIFAR-10 (Table 8) and Split CIFAR-100 (Table 9).

Table 8: Average accuracy $a(t)$ on Split CIFAR-10. We limit the learning of each task after the first one to 1 epoch.

|  | Algorithm | $a(1)$ | $a(2)$ | $a(3)$ | $a(4)$ | $a(5)$ |
|---|---|---|---|---|---|---|
| ResNet | SGD | $0.942 \pm 0.01$ | $0.777 \pm 0.15$ | $0.714 \pm 0.14$ | $0.654 \pm 0.11$ | $0.685 \pm 0.16$ |
|  | SGD$^\dagger$-10 | $0.942 \pm 0.01$ | $0.778 \pm 0.15$ | $0.713 \pm 0.16$ | $0.656 \pm 0.16$ | $0.675 \pm 0.17$ |
|  | SGD$^\dagger$-20 | $0.942 \pm 0.01$ | $\mathbf{0.779} \pm 0.16$ | $\mathbf{0.720} \pm 0.15$ | $\mathbf{0.664} \pm 0.15$ | $\mathbf{0.684} \pm 0.17$ |
| ViT | SGD | $0.903 \pm 0.003$ | $0.766 \pm 0.08$ | $0.739 \pm 0.07$ | $0.693 \pm 0.04$ | $0.7233 \pm 0.10$ |
|  | SGD$^\dagger$-10 | $0.903 \pm 0.003$ | $\mathbf{0.785} \pm 0.09$ | $\mathbf{0.751} \pm 0.09$ | $\mathbf{0.713} \pm 0.10$ | $\mathbf{0.7264} \pm 0.09$ |
|  | SGD$^\dagger$-20 | $0.903 \pm 0.003$ | $0.783 \pm 0.10$ | $0.7480 \pm 0.09$ | $0.705 \pm 0.11$ | $0.719 \pm 0.10$ |

Table 9: Final average accuracy $a(T)$ on Split CIFAR-100, compared to the average current task accuracy $a_{t,t}$ across tasks. We limit the learning of each task after the first one to 10 epochs.

|  | Algorithm | $a(T)$ | $a_{t,t}$ |
|---|---|---|---|
| ResNet | SGD | $0.2215 \pm 0.09$ | $\mathbf{0.5068} \pm 0.10$ |
|  | SGD$^\dagger$-10 | $0.2366 \pm 0.09$ | $0.4660 \pm 0.11$ |
|  | SGD$^\dagger$-20 | $0.2487 \pm 0.09$ | $0.4185 \pm 0.11$ |
|  | SGD$^\dagger$-40 | $\mathbf{0.2542} \pm 0.08$ | $0.3703 \pm 0.12$ |
| ViT | SGD | $0.2258 \pm 0.09$ | $\mathbf{0.4789} \pm 0.09$ |
|  | SGD$^\dagger$-10 | $0.2452 \pm 0.09$ | $0.4089 \pm 0.11$ |
|  | SGD$^\dagger$-20 | $\mathbf{0.2577} \pm 0.06$ | $0.3726 \pm 0.12$ |

Importantly, the average accuracy reflects the balance of forgetting and intransigence in the network. Lower forgetting comes at the cost of a lower capacity for learning new tasks. On the contrary, a very plastic network performs well on the current task while performing poorly on the old ones. In Table 9 we compare the accuracy on the current task $a_{t,t}$ to the final average accuracy of the network. SGD always outperforms SGD$^\dagger$ on the current task, however it achieves the lowest average accuracy. This compromise between forgetting and intransigence, or equivalently between plasticity and stability, is better portrayed in Figure 8, where we show the average accuracy and forgetting of SGD$^\dagger$ on Rotated-MNIST as we vary the parameter $\epsilon$, controlling the fraction of spectral energy left of the memory for each task. Simply, $\epsilon = 0$ corresponds to fixing $k = P$, and $\epsilon = 1$ corresponds to fixing $k = 0$. Importantly, the extremes achieve lower average accuracy.

### C.1.3 EXPERIMENTS SECTION 5.1

The last set of experiments is aimed at assessing the validity of the analysis.

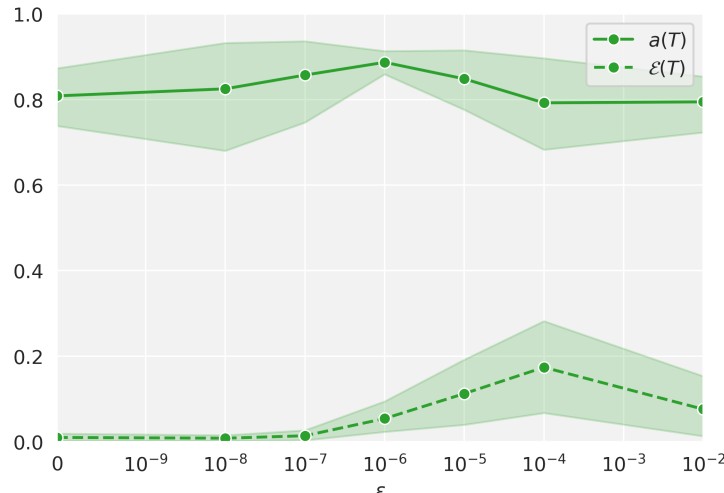

Figure 8: Average final accuracy $a(T)$ and forgetting $\mathcal{E}(T)$ of SGD$^\dagger$ on Rotated-MNIST varying $\epsilon$. For each task, a $(1-\epsilon)$ fraction of the spectral energy of the Hessian matrix is stored in memory, thus through $\epsilon$ we dynamically control the memory size.

**Approximations of forgetting**   First, we evaluate the quality of two approximations of forgetting, namely a second order Taylor approximation (Equation 2) and the approximation derived from Assumptions 1-2 (Equation 3). Table 10 extends the results shown in Table 1 to the case $o = t - 1$, which isolates the contribution of a single update to forgetting. We observe the same trend in both cases, with the second order term in the Taylor expansion providing a stable estimate of forgetting across a wider region of the parameter space. In Table 11 we report the error of the average forgetting

Table 10: Approximation error, Equation 2, on Rotated-MNIST.

|  | lr | 1$^{\text{st}}$ order | 2$^{\text{nd}}$ order | Taylor | $\mathcal{E}_o(t)$ | $\|\boldsymbol{\theta}_t - \boldsymbol{\theta}_o\|$ |
|---|---|---|---|---|---|---|
| $o \in$ | $1e^{-5}$ | $0.43 \pm 0.30$ | $0.26 \pm 0.16$ | $\mathbf{0.15} \pm 0.15$ | $0.30 \pm 0.30$ | $1.23 \pm 0.51$ |
| $[1, t-1]$ | $1e^{-2}$ | $2.33 \pm 1.87$ | $\mathbf{1.49} \pm 1.31$ | $1.55 \pm 1.32$ | $2.26 \pm 1.85$ | $8.19 \pm 2.56$ |
| $o = t - 1$ | $1e^{-5}$ | $0.16 \pm 0.07$ | $0.19 \pm 0.09$ | $\mathbf{0.05} \pm 0.02$ | $0.018 \pm 0.02$ | $0.71 \pm 0.13$ |
|  | $1e^{-2}$ | $0.39 \pm 0.06$ | $\mathbf{0.09} \pm 0.05$ | $0.13 \pm 0.05$ | $0.33 \pm 0.05$ | $5.48 \pm 0.17$ |

estimate given by Equation 3 on Rotated MNIST for different values of $t$. Interestingly, we notice that the error is not monotonic, although forgetting constantly increases.

Table 11: Equation 3 approximation error on Rotated-MNIST.

| $t$ | lr | $|\hat{\mathcal{E}}(t) - \mathcal{E}(t)|$ | $\mathcal{E}(t)$ |
|---|---|---|---|
| 2 | $1e^{-5}$ | $\mathbf{0.047} \pm 0.01$ | $0.024 \pm 0.01$ |
|  | $1e^{-2}$ | $0.062 \pm 0.07$ | $0.36 \pm 0.06$ |
| 3 | $1e^{-5}$ | $\mathbf{0.049} \pm 0.005$ | $0.122 \pm 0.02$ |
|  | $1e^{-2}$ | $0.74 \pm 0.1050$ | $1.27 \pm 0.06$ |
| 4 | $1e^{-5}$ | $\mathbf{0.073} \pm 0.02$ | $0.255 \pm 0.03$ |
|  | $1e^{-2}$ | $1.17 \pm 0.24$ | $2.45 \pm 0.23$ |
| 5 | $1e^{-5}$ | $\mathbf{0.068} \pm 0.05$ | $0.48 \pm 0.07$ |
|  | $1e^{-2}$ | $0.80 \pm 0.13$ | $3.09 \pm 0.14$ |

Finally, we evaluate the quality of the estimates of forgetting for the ResNet experiments on Split CIFAR-100. In order to do so, we employ low rank approximations of the Hessian matrix, obtained through iterative methods. In Table 12 and 13 we report results for multiple rank values, $r = 10, 20, 30, 40$. Overall, the forgetting approximation quality increases with the rank.

Table 12: Approximation error, Equation 2, on Split CIFAR-100.

| $r$ | 1$^{\text{st}}$ order | 2$^{\text{nd}}$ order | Taylor | $\mathcal{E}_o(t)$ | $\|\boldsymbol{\theta}_t - \boldsymbol{\theta}_o\|$ |
|---|---|---|---|---|---|
| 10 | $0.63 \pm 0.31$ | $0.54 \pm 0.38$ | $0.60 \pm 0.30$ | $0.57 \pm 0.39$ | $2.15 \pm 0.89$ |
| 20 | $0.63 \pm 0.31$ | $0.50 \pm 0.37$ | $0.54 \pm 0.30$ | $0.57 \pm 0.39$ | $2.15 \pm 0.89$ |
| 30 | $0.63 \pm 0.31$ | $0.41 \pm 0.30$ | $0.42 \pm 0.26$ | $0.57 \pm 0.39$ | $2.15 \pm 0.89$ |
| 40 | $0.63 \pm 0.31$ | $\mathbf{0.27} \pm 0.20$ | $\mathbf{0.21} \pm 0.16$ | $0.57 \pm 0.39$ | $2.15 \pm 0.89$ |

Table 13: Equation 3 approximation error on Split CIFAR-100.

| $r$ | $|\hat{\mathcal{E}}(t) - \mathcal{E}(t)|$ | $\mathcal{E}(t)$ |
|---|---|---|
| 10 | $0.37 \pm 0.17$ | $0.57 \pm 0.13$ |
| 20 | $0.36 \pm 0.16$ | $0.57 \pm 0.13$ |
| 30 | $0.34 \pm 0.14$ | $0.57 \pm 0.13$ |
| 40 | $\mathbf{0.30} \pm 0.11$ | $0.57 \pm 0.13$ |

**Perturbation analysis** In Section 5.1 we propose a tool to measure the range of validity of the assumptions of our analysis, named *perturbation analysis*. Here, we present the detailed results of the perturbation analysis, for all the tasks in the Rotated MNIST setting. These can be found in Figure 9.

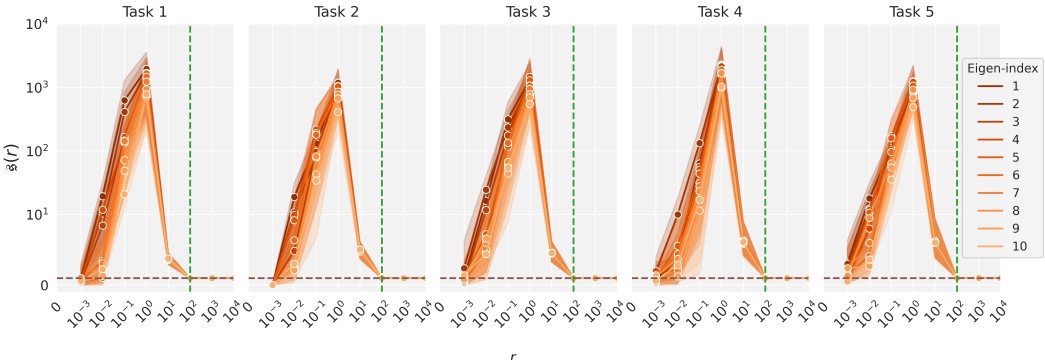

Figure 9: Perturbation score $\mathfrak{s}(r)$ on all the tasks of the Rotated-MNIST challenge, learned with SGD. The dashed green line highlights the value of r for which the score converges to 1 (marked by the dashed brown line).

Next, to further complement these results, we additionally provide results for ResNet18 in the Split-CIFAR100 setting in Figure 10 and that for ViT in Figure 11.

The observations from our perturbation analysis on Rotated-MNIST largely extend to these latter settings. Notice that, as the network size increases, the perturbation score converges to 1 for higher values of $r$ (marked by the dashed green line). There is more noise in some of the plots, which can be attributed to the fact that for these bigger networks, we are relying on the power method to obtain the top eigenvectors, unlike the exact Hessian computation in the case of Rotated-MNIST.

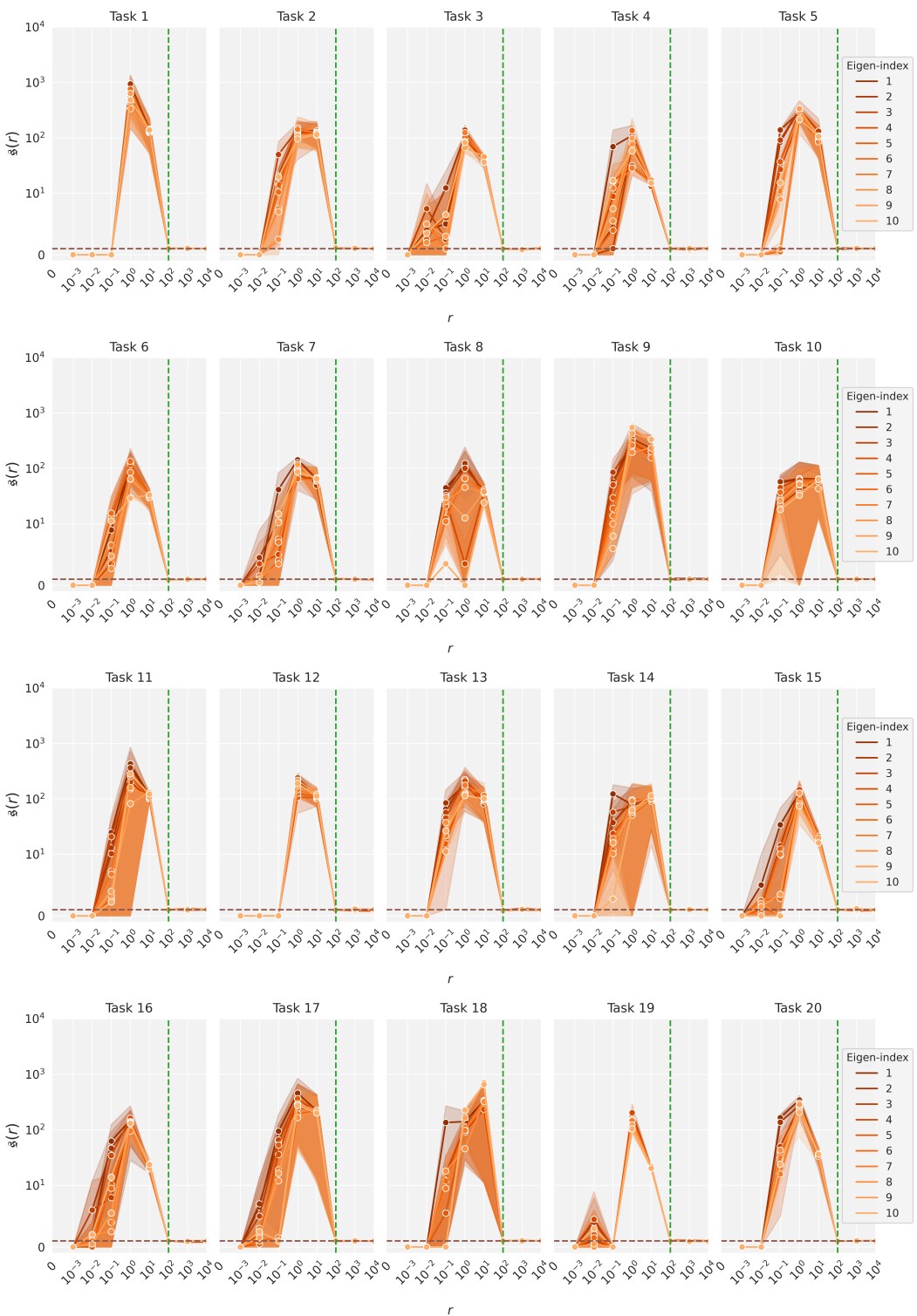

Figure 10: Perturbation score $\mathfrak{s}(r)$ on all the tasks of the Split CIFAR-100 challenge, learned with SGD. The dashed green line highlights the value of r for which the score converges to 1 (marked by the dashed brown line).

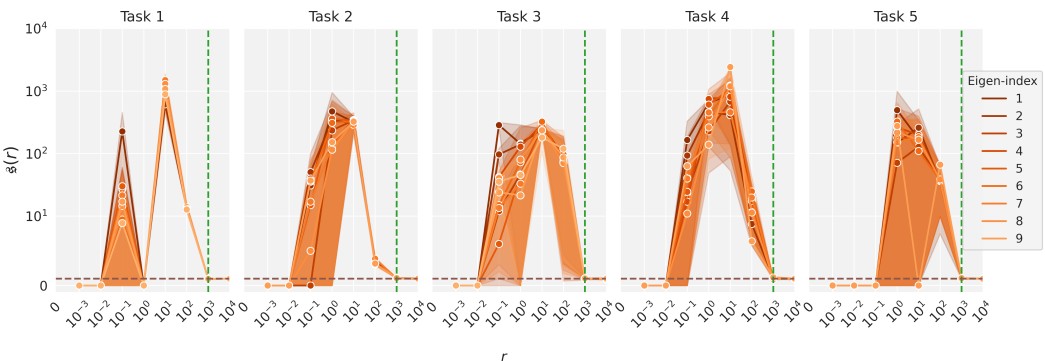

Figure 11: Perturbation score $\mathfrak{s}(r)$ on all the tasks of the Split CIFAR-10 challenge, learned with SGD. The dashed green line highlights the value of r for which the score converges to 1 (marked by the dashed brown line).

