# OpenReview forum: "Towards guarantees for parameter isolation in continual learning"
_ICLR.cc/2024/Conference — Submitted to ICLR 2024_

### Official Review · Reviewer_oq6r · 2023-10-31

**Soundness:** 2 fair
**Presentation:** 2 fair
**Contribution:** 2 fair
**Rating:** 5
**Confidence:** 3

**Summary:**

While a lot of empirical studies have been proposed to address catastrophic forgetting in continual learning, the theoretical investigation behind continual learning is quite limited. This paper studies parameter isolation based continual learning algorithms through the geometry of neural network loss landscape, and also theoretically characterizes the conditions for zero forgetting under certain assumptions. The theoretical guarantees of zero forgetting for several existing continual learning algorithms  are analyzed by using the framework proposed in this paper. Experimental studies are also conducted to justify the theoretical findings.

**Strengths:**

1. Considering that the theoretical studies are still lacking for continual learning, investigating the theoretical guarantees for continual learning algorithms is a very important direction in the community.

2. Analyzing existing orthogonal-projection based CL studies, such as OGD and GPM, from the perspective of parameter isolation methods is interesting.

3. The authors also conduct experiments to further backup their theoretical findings.

**Weaknesses:**

1. My major concern is about the definition of forgetting in equation (1), which is defined based on the training loss. However, in practice and in most empirical studies we are mainly concerned about the testing loss. This will make the theoretical results less interesting and also prevent the analysis of positive backward transfer.

2. Another concern is about the implications of the theoretical results. Based on the theoretical results, the authors further propose an augmented version of SGD. But the effectiveness of this inspired design is only justified by comparing with vanilla SGD. Note that the theoretical investigations are usually grounded by some restrictive assumptions and particularly the results in this paper are about identifying conditions for zero forgetting. If the advantage of the inspired algorithm cannot be sufficiently verified, it is hard to tell the usefulness of the theoretical findings.

**Questions:**

Besides the weaknesses above, I also have several questions:

1. The authors claimed that minimizing the average forgetting is equivalent to minimizing the multi-task loss in section 1.1. I was wondering whether this is true. In the authors' claim, the term $L_t^*$ is treated as a constant that is independent with $\theta$. But in continual learning, because the learnt model of previous tasks will be used as the initial model for current task $t$, $\theta_t$ should be correlated with  $L_{i}^*$ for any previous task $i<t$.

2. How do you define the task solutions in Assumption 2?

3. Table 1 is difficult to understand. In what scale will the errors be considered as small?


================**Post Rebuttal**================

I appreciate the authors' rebuttal. However, without clearly demonstrating that better algorithms can be developed from the theoretical results, it is hard to convince the readers that the theoretical findings are very meaningful. I will keep my rating and suggest the authors rethinking about the practical implications of the theory.

---

> ### Author Response · Authors · 2023-11-16
> **Answer to Reviewer oq6r**
>
> We are glad to hear you found our paper interesting. Hopefully, we can clarify your doubts with this answer, and improve your judgment of our work.
>
> &nbsp;
> - **"My major concern is about the definition of forgetting in equation (1), which is defined based on the training loss. "**
>
>     You raise a good point regarding the test loss. It is true that we establish guarantees for the training loss and, if the test loss significantly differs from the training loss, our guarantees are completely useless. However, as we know from statistical learning theory, the training loss converges to the test loss as the dataset size increases (keeping the model size fixed), therefore we can expect the train-test difference to be small for large enough datasets. And that is what we see in practice, for in all our experiments we measure forgetting on the test data (Section 5, Experimental Setup).
>
> &nbsp;
> - **"Another concern is about the implications of the theoretical results. Based on the theoretical results, the authors further propose an augmented version of SGD. But the effectiveness of this inspired design is only justified by comparing with vanilla SGD."**
>
>     Regarding the augmented version of SGD, SGD$^\dagger$, we have carefully stressed in the paper that we do not wish to contribute a new CL algorithm, and we only resort to the use of this modified SGD as a tool to test our theory. SGD$^\dagger$ has many problems. We cite from Section 5.3:
>     > “Overall, this training setup is suboptimal in many respects (performance, computation, memory) and it is designed specifically to validate the theoretical findings. Examples of efficient algorithms implementing the null-forgetting constraint are OGD, GPM and the other algorithms discussed, and competing with existing algorithms is not a goal of this study.”
>
>     Nonetheless, we believe that *our contribution has a value in that it progresses the theoretical understanding of continual learning algorithms, in a field especially dominated by empirical research*. Our theoretical findings not only provide **a unifying description of a set of otherwise distinct algorithms**, showing that they are essentially equivalent (and abstraction is the goal of any theory), but **it can guide principled development of new algorithms**, perhaps more efficient and effective than the existing ones.
>
>
> &nbsp;
> - (Question 1) **"The authors claimed that minimizing the average forgetting is equivalent to minimizing the multi-task loss in section 1.1 "**
>
>     In the paper we say “*The goal of continual learning algorithms is to minimise $\mathcal{E}(t)$ while learning a new task $T_t$, or, equivalently, to minimise the multi-task loss*”. Importantly ‘**while learning a new task $T_t$**’, which means in formula minimising $\mathcal{E}(t) + L_t(\theta)$. Equation (1) is a straightforward definition of forgetting and we’re not the first to use it. It’s not clear in this setting what you mean by correlation or independence.
>
>
> &nbsp;
> - (Question 2) **"How do you define the task solutions in Assumption 2?"**
>
>     The “task solutions” are simply local minima of the task loss (as said in the sentences before the Assumptions).
>
> &nbsp;
> - (Question 3) **"Table 1 is difficult to understand. In what scale will the errors be considered as small?"**
>
>     We agree, it is confusing when the forgetting value is itself very small. In any case we would take $\mathcal{E}_o(t)$ as a scale reference there.

---

### Official Review · Reviewer_5dAQ · 2023-10-31

**Soundness:** 1 poor
**Presentation:** 2 fair
**Contribution:** 2 fair
**Rating:** 1
**Confidence:** 5

**Summary:**

This paper looks at some parameter isolation methods for continual learning (including methods like OGD and GPM). It looks at conditions where forgetting is minimised or controllable, by making a 2nd order Taylor series expansion around the converged loss. Assuming that parameters do not change too much during training (ie that the Taylor series expansion is valid), the paper shows that parameter isolation methods satisfy some equations that imply zero forgetting. In experiments, the paper looks at whether the Taylor series expansion assumption holds in practice, and at the forgetting values of some algorithms (including their own algorithm, orthogonal SGD).

**Strengths:**

1. It is interesting to try and theoretically explain why parameter isolation methods may work (or not work) well. Past works have focussed mostly on regularisation and memory based methods.

2. As far as I can tell (aside from one small question, listed later), the derivations of the theorems are correct / intuitively make sense to me.

3. I found the perturbation analysis technique in Section 5.1 interesting.

**Weaknesses:**

1. At the end of Section 1.1, the paper claims that minimising average forgetting is equivalent to minimising the multi-task loss. I do not think this is true: there is a missing L_t(\theta), ie the loss at the current task t. I don't think this was used anywhere, so this should be an easy fix. However, in Section 5, the authors say, "we report forgetting in terms of accuracy", which I did not understand. What does this mean (because forgetting is not equal to accuracy)?

2. I'm not sure if OGD and GPM are 'parameter isolation' methods, a claim that seems central to the title and framing of this paper. Previous works like de Lange et al. would characterise these methods as 'constrained replay-based methods'. I do follow the authors' argument in Section 2 that these methods are not allowing parameters to be part of the parameter subspace. If anything, I would say that these methods are a combination of replay and parameter isolation methods (or, if strictly choosing one, it would be replay methods, like in de Lange it al.).

3. Assumption 2 (combined with assumption 1) seems like a very strong assumption to me. The fact that a method like OGD does not have zero forgetting on CL benchmarks (see results in other papers: OGD is not perfect!) is proof to me that assumption 2 does not hold in practice. I was very unconvinced by the experiments in Section 5.1, which looks at whether assumption 2 holds.
- In the first part (Table 1) the authors do not train until (close to) convergence, hence breaking assumption 1. It is unsurprising that training for a fixed number of epochs with a learning rate that is orders of magnitude lower leads to the parameters changing less, and hence the Taylor series expansion being more predictive (that being said, if one looks at relative values of the Taylor series expansion and E_0(t): E_0(t) is about twice the value of the Taylor value, but always within what are very large standard deviations, making this result very difficult to draw any conclusions from).
- In the second part (Figure 2), I do not understand what reasonable values of the score or of r are. Is a score of 1 good? What does an r value of 100 mean? Is this small or large? I do not know how to interpret these results.

4. Throughout the experimental section, the authors appear to break assumption 1, hence breaking the theory (at least, for small learning rates; at large learning rates, it appears that assumption 2 does not hold). This is a major problem for me.
- Also, all the error bars in Table 2 are very large, making any conclusions here hard to see.

5. If I understand correctly, the theory for OGD and GPM requires storing all input data from past tasks and constraining optimisation over this? This is a very strong requirement (and is obviously not allowed in practice in CL). This needs to be made clear, and, in my mind, reduces the significance of this result.

**Questions:**

Please see Weaknesses section.

Other minor questions / points:
1. In the proof of Theorem 1, the authors say that \Delta_2^T H_1* \Delta_2 = 0 implies that \Delta_2^T H_1 = 0. It was not clear to me why this is the case, can the authors please expand on this?
2. Typo in the line after Equation 3: I believe it should be \Sum_{o=1}^{t-1}, currently it says t-2.

---

> ### Author Response · Authors · 2023-11-16
> **Answer to Reviewer 5dAQ**
>
> We are quite surprised by your score and hope, despite the high confidence, that we can convince you to reconsider with this answer.
>
> ### Weaknesses
> - "**At the end of Section 1.1, the paper claims that minimising average forgetting is equivalent to minimising the multi-task loss.**"
>
>    Citing from the paper, we say “*The goal of continual learning algorithms is to minimise $\mathcal{E}(t)$ **while learning a new task** $\mathcal{T}_t$ or equivalently to minimise the multi-task loss*.” If you are referring to this sentence, perhaps you might have missed the ‘*while learning a new task*’ part, which is exactly what you claim to be missing?
>
> &nbsp;
> - **"In Section 5, the authors say, "we report forgetting in terms of accuracy", which I did not understand [...] What does this mean (because forgetting is not equal to accuracy)?"**
>
>     Yes, forgetting is not equal to accuracy. As defined in Equation (1), forgetting is the difference in loss after a change in parameters. However *the loss is often not easy to interpret, so we instead quantify the results in terms of difference in accuracy after a change in parameters*. We should clarify this better in the paper. Note that a precise definition has been given in the Appendix (Section B.0.1, Experimental Setup - Metrics).
>
> &nbsp;
> - **"I'm not sure if OGD and GPM are 'parameter isolation' methods, a claim that seems central to the title and framing of this paper."**
>
>    - You’re right in noticing that our definition of parameter isolation algorithms is wider than the convention in the literature. This is a point we should stress more in the paper. In a way, *we are proposing a change in the vocabulary, and we do so based on the results of our analysis*.
>    - *We define parameter isolation as algorithms partitioning the network parameters between tasks*, and we show that OGD and GPM (and consequently all their variants) do it in a more sophisticated way than PackNet or Progressive Networks.
>    - Essentially, **we show that all these algorithms rely on the same mechanism** (i.e. tasks parameters orthogonality) and therefore they can be seen as special cases of a unique strategy. Similarly, all regularization-based methods rely on soft constraints on the parameter change (determined by the regulariser) and replay-based methods rely on empirical, small-sample approximations of the previous tasks' losses.
>    - Therefore *labeling OGD and GPM (and their variants) as replay methods does not really reflect the nature of these algorithms*, which is much more similar to that of PackNet than Experience Replay. However **the connection to parameter isolation has not been ascertained before in the literature**, and we accomplish this with Theorems 3-5 in this paper.
>    - Moreover, several different categorizations of continual learning algorithms exist today, each of them using different terms. *One may say that the continual learning terminology is for most parts still in progress, and should not be regarded as final yet.*
>
> &nbsp;
> - **"Assumption 2 (combined with assumption 1) seems like a very strong assumption to me."**
>
>    It is. We do not claim that OGD is perfect, and we are well aware that it does not achieve zero forgetting in practice. **Our theory can explain why that happens**, which is precisely when Assumption 2 is violated. In fact, we even provide experimental data (Figure 3) to support this claim.
>
>     In Section 5.1 we ask at what point this assumption breaks and we do not conclude that it always holds (which is naturally unrealistic). Additionally, *we would like to highlight that later in the paper we do extend our theory to settings where we can drop Assumption 2*, like GPM. Therefore **our theory is not conceptually limited by the assumption**.
>
> &nbsp;
> - **"In the first part (Table 1) the authors do not train until (close to) convergence, hence breaking assumption 1."**
>
>      In our first set of experiments our goal is to assess Assumption 2. Note that *we do not need the gradients to be zero, i.e. Assumption 1, for the second order Taylor expansion* (Equation 2). Therefore, **it is not necessary in this case to train to convergence**. Regarding the large standard deviations, that is the effect of using different seeds. As you might guess, the initialisation can play an important role in the geometry of the loss around it.
>
> &nbsp;
> - **"In the second part (Figure 2), I do not understand what reasonable values of the score or of r are."**
>
>     We have expanded on the interpretation of $s(r)$ in the answer to Reviewer W8iZ. To give you a short answer, a score $s(r)=1$  means that moving along an eigenvector, by a distance $r$,  has the same effect as moving along a random other direction, by the same distance $r$, in the parameter space. Consequently, the values of $r$ for which the score is 1 identify regions of the parameter space where the quadratic approximation of the loss does not hold anymore.
>
>
> (continued in the following comment)

---

> > ### Author Response · Authors · 2023-11-16
> > **Answer to Reviewer 5dAQ (part 2)**
> >
> > - **"Throughout the experimental section, the authors appear to break assumption 1, hence breaking the theory."**
> >
> >     In Assumption 1 we do not require the gradient norm to be 0, but simply to be small ($\epsilon$). All our approximations hold with an error in the order of $\epsilon \cdot \delta$, which is good enough for most of the cases. The experimental results (Sections 5.2 and 5.3) arguably demonstrate that Theorem 1 and 2 are  robust to Assumption 1 violations.
> >
> > &nbsp;
> > - **"Also, all the error bars in Table 2 are very large, making any conclusions here hard to see."**
> >
> >     In Appendix C.1.1 we show the same results in a plot (Figure 6 and 7). It might be easier to see a trend there. In general, the high error bars are a result of the instability of iterative eigen-decomposition, which is the only possible way to access the eigenspaces of Hessian matrices of larger networks.
> >
> > &nbsp;
> > - **"If I understand correctly, the theory for OGD and GPM requires storing all input data from past tasks and constraining optimisation over this?"**
> >
> >      In principle yes, but in practice, in the respective papers, the authors show that a couple hundred examples randomly sampled from the training set is usually enough to get a good approximation. Anyway, OGD and GPM (and the variants that have followed) are not of our making. Our contribution concerns itself with explaining the inner workings of parameter isolation algorithms (to which, we argue, OGD and GPM belong).
> >
> > &nbsp;
> > ## Questions:
> >
> > - ***"In the proof of Theorem 1, the authors say that \Delta_2^T H_1* \Delta_2 = 0 implies that \Delta_2^T H_1 = 0. It was not clear to me why this is the case, can the authors please expand on this?"**
> >
> >      That is a simple fact of linear algebra. It’s easy to see once you write out the product $v^\intercal M v$ as $\sum_i \sigma^M_i \cdot \langle v, v^M_i\rangle^2$, where $\sigma_i^M$, $v^M_i$ are the eigenvalues and eigenvectors of the matrix $M$, and similarly write $v^\intercal M$ as $  \sum_i \sigma^M_i \cdot  \langle v, v^M_i\rangle \cdot v^M_i$. Setting the first one to $0$ will imply, given that the matrix M is positive semi definite, that all the elements in the sum are $0$. Hence the second term will be a $0$ vector.
> >
> > &nbsp;
> > - **"Typo in the line after Equation 3: I believe it should be \Sum_{o=1}^{t-1}, currently it says t-2."**
> >
> >     It’s not a typo, the two formulations are equivalent (just notice that for $o=t-1$ the term cancels).

---

> > ### Comment · Reviewer_5dAQ · 2023-11-16
> > **Response**
> >
> > Thanks to the authors for their rebuttal. I will go through the points using the same numbers as in my original review.
> >
> > 1. Thank you for pointing this out. I think I read the 'while' as a 'when'. I see why what you have written makes sense. I request you make it clearer so others do not also get confused, but I agree this is not a negative point of the paper.
> >
> > 2. I think it is very important to stress this as a contribution of your work, to the extent that this should probably go in your abstract and more details in the introduction. I agree with you that different works categorise the literature differently, and this is evolving. However, the works that use this terminology do not define parameter isolation methods like you do. Your definition is perfectly reasonable, and I see the logic. But the claims you make in the abstract are then misleading, as I (the reader) think I already know what 'parameter isolation' methods means, when you are in reality redefining it.
> >
> > 3. (and 4.) Perhaps I should say, when there is a low learning rate and fixed number of epochs, then it is unsurprising that the parameters change very much, and therefore that the approximation is better. But I would also assume that the overall average accuracy of the algorithm would correspondingly be lower, as you are not training to convergence on every task (ie break Assumption 1 / have very large epsilon values), and therefore do not learn the new tasks well. Is this true for all experiments (including ones in eg Table 2)? I found some final average accuracies in Appendix C.1.2 but these seem quite low, especially for Split CIFAR100.
> >
> > - Thank you for explaining meanings behind s(r) and r.
> > - Looking at Figs 6 and 7, the error bars seem much lower than in Table 2. I think I'm missing something, can the authors please clarify?
> > - Unfortunately the large error bars make conclusions very, very difficult. This implies to me that, based on randomness, there are many times when one method is much better than another. For example, saying something like "the effect is stronger when using 20 instead of 10 dimensions" is not clear to me as everything is so marginal.
> >
> > 5. It's important to note that your theory assumes access to all past data (please correct me if I'm wrong), which these methods in practice obviously do not have.
> >
> > Questions: thank you for clarifying both these points, what you say makes sense.
> >
> > At this stage, I am considering increasing my score to 3, however, for reasons listed above (especially the point on small learning rates), I do not think this is ready for publication.

---

> > > ### Author Response · Authors · 2023-11-19
> > > **Answer to Reviewer 5dAQ**
> > >
> > > Thank you for the quick response. We posted the rebuttals as soon as we could with the hope to engage in a constructive discussion with the reviewers. Let us elaborate further on some of the points above.
> > >
> > > &nbsp;
> > > - (2) “**I think it is very important to stress this as a contribution of your work, to the extent that this should probably go in your abstract and more details in the introduction.**”
> > >
> > >    We understand your point of view, and we think you raise a valid point. **We can change the abstract and introduction to present our definition of parameter isolation**. In the current version of the paper, we discuss it in the Related Work section.
> > >
> > > &nbsp;
> > >
> > > -  (3-4) “**(and 4.) Perhaps I should say, when there is a low learning rate and fixed number of epochs, then it is unsurprising that the parameters change very much, and therefore that the approximation is better. But I would also assume that the overall average accuracy of the algorithm would correspondingly be lower [...]**”
> > >
> > >     As stated in the introduction of Section 5, the aim of our experiments is to test the theory. With the last experiment, where we try out different architectures, **we want to make the point that our theory is agnostic to the specific parametrization (e.g. architecture) of the network**. We believe that its generality is an important attribute of our analysis. Consequently, we design the experiment to best approximate a setting where the assumptions would be (both) satisfied, regardless of the final performance. Notice that, *although the absolute value of the average accuracy is low for CIFAR100, the highest value is achieved when virtually partitioning the network with SGD$^\dagger$*. This is a positive result in our view. One must also consider that *CIFAR100 is a particularly difficult challenge for this setting, since it’s a sequence of 20 tasks*, and thus Assumption 2 will likely be violated. **The performance on CIFAR10, for example, is much higher** (these results are also in Appendix - Section C.1).
> > >
> > > &nbsp;
> > >
> > > -  “**Looking at Figs 6 and 7, the error bars seem much lower than in Table 2. I think I'm missing something, can the authors please clarify?**”
> > >
> > >    Thanks for pointing this discrepancy out! Looking more closely into this issue, **we have realized that there was a mistake in the code producing the results**. Basically, instead of computing the average forgetting $\mathcal{E}(t) = \frac{1}{t} \sum\_{o} \mathcal{E}\_o(t)$ for each seed $s$ (say $\mathcal{E}\_s(t)$) and then calculating the standard deviation, we directly computed the standard deviation on the average of $\frac{1}{s} \frac{1}{t} \sum_{s} \sum_{o} \mathcal{E}\_{t,s}(o)$. This mistake inflated the standard deviation values. Similarly for the Taylor approximation experiments. **We corrected Table 1 and Table 2 accordingly and updated the paper** (see new version attached), which now should look much better.
> > >
> > > &nbsp;
> > > - “**Unfortunately the large error bars make conclusions very, very difficult.**”
> > >
> > >      Given the previous point, we believe this should be fixed now!
> > >
> > > &nbsp;
> > > - “**It's important to note that your theory assumes access to all past data (please correct me if I'm wrong), which these methods in practice obviously do not have.**”
> > >
> > >     That is true. It is common in machine learning to provide analyses using the entire dataset, and it often works in practice to have finite approximations using only a random subset of the dataset (think of batch gradient descent for instance). Therefore we would argue that this observation does not make our theory less useful in practice.
> > >
> > >
> > > &nbsp;
> > >
> > > We hope that with this answer, considering the empirical results improvement and the discussed changes in the presentation -point (2)-,  we have persuaded you of the value of our work. Please let us know if there are any further points of discussion that you would like us to consider.

---

### Official Review · Reviewer_W8iZ · 2023-11-01

**Soundness:** 3 good
**Presentation:** 3 good
**Contribution:** 2 fair
**Rating:** 6
**Confidence:** 1

**Summary:**

This paper conducts a second-order analysis of continual learning and derives theorems that lead to small catastrophic forgetting under mild conditions. The authors show that OGD, GPM, and explicit parameter isolation strategies can be explained using the derived theorems.

**Strengths:**

The theoretical framework is powerful enough to include many existing continual learning techniques.

**Weaknesses:**

The second-order analysis is restrictive to the case when the learning rate is small.

**Questions:**

1. On page 3, in "Catastrophic forgetting from the loss geometry viewpoint", you mention that parameter isolation strategies can provide stronger guarantees than regularization-based methods. Is the intention to stress that your analysis doesn't involve the optimization procedure? I want to confirm this since you did not discuss regularization-based methods in later sections in detail.
2. Can you also give a brief comparison between your results and the guarantees of the regularization-based methods?
3. When we talk about the strict parameter isolations, does the network know the task domain index at training and inference time?
4. Can you explain the columns in table 1? What does "Taylor" mean here?
5. In "Perturbation analysis" on page 7, you mention that it is to measure assumption 2, do you mean to verify eq(2)?
6. Can you give some intuition behind the perturbation score s(r)? Why is it a good measure of the approximation error?

---

> ### Author Response · Authors · 2023-11-16
> **Clarification and answers to Reviewer W8iZ**
>
> Thank you for your positive review and we appreciate the good quality of your questions. With the following, we hope to clarify all your earlier concerns.
>
> - **"The second-order analysis is restrictive to the case when the learning rate is small."**
>
>      A second-order Taylor expansion, like any local approximation of a function, naturally holds only within a given distance from its center.  In our work we show that **some parameter-isolation methods, like OGD, rely on a second-order approximation of the loss, and therefore are fundamentally limited by the locality of the approximation**. This is an important result in our view, because it informs us of the inherent flaws of the algorithm.  However, we can successfully apply our analysis to other algorithms such as GPM which elude the locality constraint by using different second order approximations along the optimization path (see Theorem 5 for details). **Thus the 'small learning rate' constraint is not a weakness of our analysis, rather of some of the algorithms analysed**.
>
> &nbsp;
> - (Question 1) **"You mention that parameter isolation strategies can provide stronger guarantees than regularization-based methods.  Is the intention to stress that your analysis doesn't involve the optimization procedure?"**
>
>     Yes, that is correct. By design *regularisation-based algorithms reach a compromise between the task loss and the regulariser*. However, the extent to which they compromise in favor or at odds with the regularisation ultimately depends on the optimization process. On the contrary, **parameter isolation methods enforce a hard constraint on the parameters, which holds independently of the optimization method employed**. If you will, the difference between regularisation-based and parameter isolation is akin to the difference between regression with L2 regulariser and sparse linear regression.
>
> &nbsp;
> - (Question 2) **"Can you also give a brief comparison between your results and the guarantees of the regularization-based methods?"**
>
>     That’s an important point. Yin et al. provide theoretical results akin to ours in the space of regularisation methods. They carry out an optimization-based analysis, introducing smoothness assumptions and they provide convergence results for SGD with CL regularization and a generalization result (which is akin to a guarantee of optimality) holding in probability. *In a nutshell, the regularization setup does not allow for as strong guarantees as the parameter-isolation setup* (i.e. guarantees that hold regardless of smoothness, optimization algorithm or sample set size). The reason for that is that **regularization methods enforce low forgetting as a soft constraint, while parameter isolation methods enforce a hard constraint on the parameters**.
>
>    Interestingly, our results align with those of Yin et al. for regularisation-based methods in showing that the constraint enforced by the algorithm is based on a second-order approximation of the loss. Thus **regularisation and parameter isolation methods use the same information to prevent forgetting, however the former resorts to soft while the latter to hard constraints**.
>
> &nbsp;
> - (Question 3) **"When we talk about the strict parameter isolations, does the network know the task domain index at training and inference time?"**
>     That is a good question. It ultimately depends on the specific method. For example, PackNet uses a task-specific parameter binary mask, and network expansion methods assign a different parameter set to each task. On the other hand, sparsity-based algorithms like SpaceNet do not need to know task identity information at test time, but use it while training.
>
>
> (continued in the following comment)

---

> > ### Author Response · Authors · 2023-11-16
> > **Clarification and answers to Reviewer W8iZ (part 2)**
> >
> > - (Question 4) **"Can you explain the columns in table 1? What does "Taylor" mean here?"**
> >
> >    Looks like we forgot to include a brief description of the table here, thanks for pointing this out. Taylor is simply the approximation obtained by the full second order Taylor expansion (i.e., when including the first and second order terms). In the first 3 columns we report approximation errors, while in the fourth one we report the actual forgetting value (the quantity which is being approximated).
> >
> > &nbsp;
> > - (Question 5) **"In "Perturbation analysis" on page 7, you mention that it is to measure assumption 2, do you mean to verify eq(2)?"**
> >
> >     With the perturbation analysis, we assess how much the locality constraint (so Assumption 2) can be violated before the second order approximation breaks down. In a way, you can see it as a measure of the validity of Equation (2) as a function of ‘r’, the distance from the approximation center.
> >
> > &nbsp;
> > - (Question 6) **"Can you give some intuition behind the perturbation score s(r)?"**
> >
> >      The score $s(r)$ intuitively says how much bigger is the change in the loss when we perturb along a principal eigenvector compared to the average change we would see along a random direction. *The principal eigenvectors of the Hessian correspond to the directions of the steepest change of the loss function*, according to a second order approximation around the point where you evaluate the Hessian. Too far away from this point the approximation starts to lose its validity, and the eigenvectors are not in exact correspondence with the directions of maximal change. As a consequence, the effect on the loss moving along an eigenvector far away from the approximation focal point starts becoming comparable to the effect on the loss moving along any other direction, and the score will be close to $1$. *Thus the region for which $s(r)$ is strictly higher than $1$ corresponds to the region around the minimum where a second order approximation is valid*.

---

### Official Review · Reviewer_9KMy · 2023-11-02

**Soundness:** 3 good
**Presentation:** 2 fair
**Contribution:** 2 fair
**Rating:** 5
**Confidence:** 3

**Summary:**

The authors construct a unified framework to analyze the effectiveness of parameter isolation methods in continual learning. With the 2nd order Taylor expansion on the objective function, the authors derive some necessary and sufficient conditions for guaranteed non-forgetting. They further show some existing parameter isolation methods are special cases of their framework.

**Strengths:**

- The authors construct a theoretical framework for parameter isolation strategies through the lens of loss landscape.

- Several seminal continual learning methods can be analyzed and explained with their framework.

**Weaknesses:**

- My major concern is the significance or novelty of the submission. The framework authors formulate, i.e., analyzing null forgetting in continual learning through loss landscape, is somehow not quite new given some previous works. Besides, though the authors analyze some continual learning methods with their framework, it would be more appreciated if a new continual learning method can be proposed guided by the theoretical framework. The current presentation also makes the experimental part weak, which doesn’t provide any very significant results to the field.

- The authors analyze a few continual learning methods that already have sound intuition on their effectiveness, especially the OGD and GPM. What about other more empirical methods, e.g., the prompt tuning continual learning method [1].

#### Reference:
Wang, Zifeng, et al. "Learning to prompt for continual learning." Proceedings of the IEEE/CVF Conference on Computer Vision and Pattern Recognition. 2022.

**Questions:**

What does $\xi_{\tau}(t)$ in Theorem 2 mean?

---

> ### Author Response · Authors · 2023-11-16
> **Comments and answers to Reviewer 9KMy**
>
> - "**My major concern is the significance or novelty of the submission.**"
>
>      We are certainly not the first to analyze continual learning algorithms from a geometrical perspective and we do not claim to be pioneering the tools we use, such as the Taylor expansion. However, we believe our contribution to be unique because, as far as we know, we’re the first to have shown that **a rather large number of methods (which we refer to as ‘parameter isolation’) follow the same general principle**. A similar statement has been made previously for all regularization-based algorithms, but that’s a different set of methods.
>
>     Also, we definitely agree that a favorable outcome of this line of research would be the development of principled algorithms. However, we believe that theoretical contributions such as ours are as valuable and necessary as empirical contributions, *even more so in a field like continual learning that is largely dominated by empirical methods*.
>
> &nbsp;
>
> -  "**The authors analyze a few continual learning methods that already have sound intuition on their effectiveness, especially the OGD and GPM.**"
>
>      Our theory applies to any method which, implicitly or explicitly, partitions the parameter space between tasks, e.g. through a linear projection (OGD), parameter freezing (PackNet), sparsity constraints, etc… So *it could in principle also be applied to Wang et al., if the parameters used to learn the sequence of tasks were also partitioned*. We do not cover all the algorithms in the parameter isolation family in our paper, because that would be unfeasible within the space limits, and it would be pointless given that new algorithms are proposed all the time. However, *the examples we propose are meant to provide an intuition of how to apply the same strategy to other methods*. In particular, **for OGD and GPM (although they are ‘intuitively sound’) the connection to parameter isolation is not so clear**, as shown by the extensive proofs backing the result. On the other hand, for ‘strict parameter isolation’ methods (Section 4.3) we do not go into the proof detail, because the link to parameter isolation is obvious. We would like to stress that in the respective papers, the authors of OGD and GPM do not establish any theoretical guarantees for the effectiveness of their methods, and that we are the first to obtain them.
>
> &nbsp;
>
> - Regarding your question: *$\mathcal{E}\_\tau(t) $ , is the forgetting of task $\tau$ after learning task $t$* (see Section 1.1, after Equation (2)). In the specific case of Theorem 2, we identify a constraint by which $\mathcal{E}\_\tau(t)=0$ for all tasks $\tau<t$.
>
> We hope to have addressed all your concerns and to have better communicated the contribution of our work so as to improve your judgement of the paper. If you have any new questions please let us know, we’re happy to answer them.

---

### Author Response · Authors · 2023-11-19
**New version of the paper**

Dear all reviewers and AC,

In dialogue with one reviewer we have realised an important mistake in the calculation of our results which inflated the standard deviation values, weakening our empirical evidence. We have now fixed the error and updated a new version of the paper, with correct (and much smaller) standard deviation. In the interest of time, we have not yet applied these changes to the results in the appendix. The changes would definitely be finalised for the camera ready.

We encourage all reviewers to re-evaluate our work, given the new paper revision, and the arguments we have presented in the rebuttal discussions. Moreover, if there were further points of criticism towards our work, we would like to encourage the reviewers to engage in a discussion with us, and expose these points, since there is still some time before the end of the rebuttal period.

Thank you,
The authors

---

### Author Response · Authors · 2023-11-22
**Invitation to reply**

As we are approaching the close of the discussion period, we would like to make one last effort in engaging with the reviewers. To reiterate, we have incorporated the reviewer feedback and revised our paper. We  believe that we have adequately addressed all the raised concerns.

&nbsp;

*In light of these changes and clarifications, we ask that the reviewers, especially those who had a negative initial impression of our paper, to reevaluate their stance.* We understand the burden of reviewing, however the success of the peer-review system hinges on a active dialogue between the involved parties, and we sincerely hope for your engagement.

---

### Meta-Review · Area_Chair_oEtp · 2023-12-03

**Metareview:**

**Summary:**

The paper presents a theoretical investigation into parameter isolation methods in continual learning, focusing on the geometry of neural network loss landscapes. It aims to establish theoretical guarantees against catastrophic forgetting. The authors propose a unified framework to analyze and understand several existing continual learning algorithms, particularly those based on parameter isolation strategies. They utilize a second-order Taylor expansion to derive conditions for minimal or zero forgetting. The paper also includes experimental studies to validate these theoretical findings.

**Strengths:**
1. The paper contributes to the limited theoretical understanding of continual learning algorithms, a domain predominantly driven by empirical research.

2. The approach to analyze existing algorithms like OGD and GPM under a common framework of parameter isolation is noteworthy.

3. The authors attempt to back their theoretical claims with experimental data, which is commendable for a study of this nature.

**Weaknesses:**
1. The paper defines forgetting based on training loss, which may not accurately reflect practical scenarios where test loss is more relevant. This gap potentially limits the applicability of the theoretical insights.

2. Despite the theoretical advancements, the paper falls short in demonstrating the practical utility of these findings. The augmented version of SGD proposed in the paper, while interesting, is not convincingly shown to be superior to existing methods in practical settings.

3. Some of the assumptions made, such as the small learning rate and fixed epochs, restrict the generalizability of the results. The experimental validation, while present, is not sufficiently robust to convincingly support the theoretical claims, especially considering the large error bars and the limitations in the experimental design.

**Justification For Why Not Higher Score:**

In conclusion, while the paper addresses an important theoretical gap in continual learning research, the concerns regarding the practical significance of the findings, coupled with the limitations in experimental validation and clarity, suggest that it may not be ready for publication in its current form.

**Justification For Why Not Lower Score:**

N/A

---

### Decision · Program_Chairs · 2024-01-16

Reject